# VAP spatially stabilizes dendritic mitochondria to locally support synaptic plasticity

Ojasee Bapat [1,2], Tejas Purimetla[1,5], Sarah Kruessel [3,6], Monil Shah [1,2], Ruolin Fan[1], Christina Thum[3], Fiona Rupprecht[3,4,7], Julian D. Langer[3,4] & Vidhya Rangaraju [1] ✉

Synapses are pivotal sites of plasticity and memory formation. Consequently, synapses are energy consumption hotspots susceptible to dysfunction when their energy supplies are perturbed. Mitochondria are stabilized near synapses via the cytoskeleton and provide the local energy required for synaptic plasticity. However, the mechanisms that tether and stabilize mitochondria to support synaptic plasticity are unknown. We identified proteins exclusively tethering mitochondria to actin near postsynaptic spines. We find that VAP, the vesicle-associated membrane protein-associated protein implicated in amyotrophic lateral sclerosis, stabilizes mitochondria via actin near the spines. To test if the VAP-dependent stable mitochondrial compartments can locally support synaptic plasticity, we used two-photon glutamate uncaging for spine plasticity induction and investigated the induced and adjacent uninduced spines. We find VAP functions as a spatial stabilizer of mitochondrial compartments for up to ~60 min and as a spatial ruler determining the ~30 μm dendritic segment supported during synaptic plasticity.

Neurons have a complex morphology where most synapses are far from the cell body. With increasing awareness of local processing at synapses[1], it is clear that synapses are hotspots of energy consumption. Therefore, a local energy source is necessary to cope with the fast and sustained energy demands of synapses[2–7], as mere ATP diffusion from the cell body is insufficient[8]. Consistent with this notion, we recently showed that mitochondria form unique temporally (60–120 min) and spatially (30 μm) stable compartments by tethering to the cytoskeleton, actin and microtubule, in dendrites[6]. The definition of a mitochondrial compartment, comprising single or multiple mitochondria, is where a sub-region of a compartment shows continuity with the rest of the compartment in fluorescence loss in photobleaching and photoactivation experiments[6]. Notably, we found that perturbation of a dendritic mitochondrial compartment function abolishes protein synthesis-dependent plasticity in spines[6]. Therefore, these stable mitochondrial compartments serve as local energy supplies to fuel synaptic plasticity in dendrites. What remains unknown is the mechanism that stabilizes mitochondria locally in dendrites and enables synaptic plasticity.

In contrast to dendritic mitochondria, most mitochondria are motile in axons within the long durations (minutes to hours) of plasticity formation and maintenance[6]. Interestingly, when the cytoskeleton (actin or microtubule) is depolymerized, stable dendritic mitochondria get destabilized, but in contrast, the motile axonal mitochondria get stabilized[6,9]. This observation suggests that dendritic mitochondria interact differently with the cytoskeleton than axonal mitochondria. This mechanism might be facilitated by exclusive protein(s) tethering mitochondria to the cytoskeleton in

[1]Max Planck Florida Institute for Neuroscience, Jupiter, FL 33458, USA. [2]International Max Planck Research School for Synapses and Circuits, Jupiter, FL 33458, USA. [3]Max Planck Institute for Brain Research, Frankfurt 60438, Germany. [4]Max Planck Institute of Biophysics, Frankfurt 60438, Germany. [5]Present address: Geisel School of Medicine at Dartmouth, Hanover, NH 03755-1404, USA. [6]Present address: Johns Hopkins University School of Medicine, Baltimore, MD 21205, USA. [7]Present address: Thermo Fisher Diagnostics GmbH, Henningsdorf 16761, Germany. ✉e-mail: vidhya.rangaraju@mpfi.org

dendrites that are either absent in axons or not associated with axonal mitochondria.

Many proteins and their mechanisms involved in docking axonal mitochondria have been well-studied near axonal terminals and during axon branching. Notably, most of these studies have focused on mitochondrial-microtubule interactions[10,11,12] and a few on mitochondrial-actin interactions[13,14]. However, the mechanisms required for stabilizing dendritic mitochondria near postsynaptic spines, particularly to support synaptic plasticity, remain largely unexplored[15]. Given that dendrites and axons are biochemically, structurally, and functionally distinct, and the mitochondria in these two compartments also behave very differently[6,16–18], it is essential to characterize the specialized mitochondrial mechanisms in dendrites and how they enable synaptic plasticity.

Here, we set out to find the protein tether required for mitochondrial-cytoskeletal interaction and if it determines the spatial size of the dendritic segment supported during synaptic plasticity. For instance, the length of the mitochondrial compartment (30 μm) might determine the spatial spread and availability of ATP within a dendritic segment. Such a spatial organization might be necessary for clustered synaptic plasticity, where a cluster of neighboring spines within a certain distance from a plasticity-induced spine exhibit plasticity[1,19]. As clustered plasticity is important during learning, experience, and development[20–25], it is essential to understand the underlying molecular mechanisms that support them.

Using advances in proximity-based mitochondrial proteome labeling and imaging mitochondrial-actin tethering and stabilization[6,26–28], we have identified proteins tethering mitochondria to actin exclusively in

dendrites. In addition, we find that the vesicle-associated membrane protein-associated protein (VAP)[29–31] plays a central role in stabilizing dendritic mitochondria for long durations of synaptic plasticity formation and maintenance and determining the spatial segment supported during synaptic plasticity.

## Results

### Proximity labeling strategy to label mitochondria interactors

To identify the proteins spatially stabilizing dendritic mitochondria, we first screened for proteins required for mitochondrial-cytoskeletal interaction in dendrites. We employed a proximity labeling strategy based on the enzyme APEX2, an engineered form of soybean ascorbate peroxidase[32]. APEX2 can catalyze the promiscuous biotinylation of proteins within a few nanometers of its active site[32]. Most studies with APEX2 have been done in either non-neuronal cells or non-hippocampal neurons. As we are interested in studying synaptic plasticity in hippocampal dendrites, we first validated and characterized APEX2 biotinylation of proteins in dendrites of primary hippocampal neuronal cultures. We expressed APEX2 on the outer mitochondrial membrane (APEX-OMM, APEX2 is simplified as APEX in the rest of the text) and labeled the neurons with biotin phenol and hydrogen peroxide (Fig. 1a, see Methods)[26]. APEX-OMM expression colocalized with the spatially confined mitochondrial signal (visualized by EGFP targeted to mitochondria) in neurons (Supplementary Fig. 1a–c). However, compared to the spatially confined expression of APEX-OMM, the biotinylation signal of APEX-OMM was cytosolically distributed, indicating the diffusion of the biotinylated proteins from the mitochondrial membrane surface to the neighboring neuronal

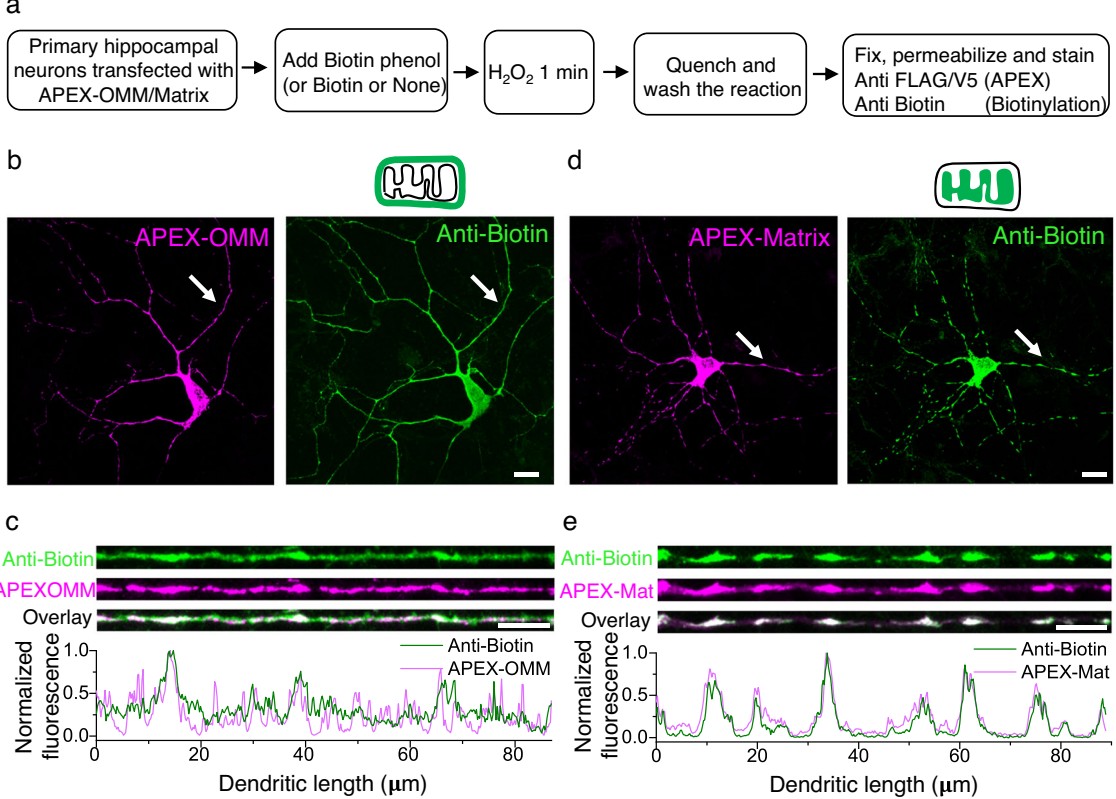

**Fig. 1 | APEX strategy to label proteins interacting with neuronal mitochondria.** **a** Experimental workflow for APEX-based proteome labeling of the mitochondrial outer membrane (OMM) and mitochondrial matrix. **b** Representative images showing APEX expression on the OMM (magenta) and the biotinylated proteome (green). Total n: 9 neurons, 2 animals. Scale bar, 20 μm. **c** Line profiles of respective dendrites pointed in (**b**) (white arrows) show diffused biotin labeling with APEX-OMM. Scale bar 10 μm. **d** Representative images showing APEX expression within the matrix (magenta) and the biotinylated proteome (green). Scale bar, 20 μm. Total *n*: 60 neurons, 5 animals. **e** Line profiles of the respective dendrites pointed in (**d**) (white arrows) show spatially confined biotin labeling compared to APEX-OMM in (**c**). Scale bar 10 μm. Source Data files are provided.

regions (Fig. 1b, c). As a Control, we expressed APEX targeted to the mitochondrial matrix (APEX-Matrix)[32]. The APEX-Matrix expressing neurons exhibited a spatially-confined biotin signal overlapping with the spatially confined APEX-Matrix expression, indicating the compartmentalization of biotinylated proteins within the mitochondrial matrix (Fig. 1d, e). Furthermore, biotin labeling was not observed in the absence of biotin phenol or the presence of an alternative substrate, biotin (Supplementary Fig. 1d, e), suggesting that biotinylation of proteins is specific to the presence of biotin phenol as the substrate. These experiments confirmed that APEX-OMM-based proximity labeling could be used to biotinylate proteins present on and interacting with the mitochondrial outer membrane in hippocampal neurons.

### Identifying interactors of the OMM and actin

To isolate proteins interacting with the mitochondrial outer membrane, we expressed primary hippocampal neuronal cultures with APEX-OMM,

labeled the neurons with biotin phenol and hydrogen peroxide, enriched the biotinylated proteins using streptavidin, digested, and analyzed them using liquid chromatography-coupled tandem mass spectrometry (LC-MS) (Fig. 2a). Neurons labeled either in the absence of APEX-OMM or hydrogen peroxide were used as negative controls (Fig. 2a–c). Western blot analysis of the streptavidin-enriched proteins showed biotinylated proteins in APEX-OMM samples compared to the negative control without APEX-OMM (Supplementary Fig. 2a); with an equal amount of total protein loaded in both conditions confirmed by Coomassie stain (Supplementary Fig. 2b).

An exclusive APEX-OMM proteome list was obtained from three replicates after subtracting proteins found in the negative controls without APEX-OMM and hydrogen peroxide (Fig. 2b, representative replicate). Next, we consolidated a list of exclusive APEX-OMM proteins found in at least two replicates, resulting in 129 proteins (Fig. 2c, Supplementary Table 1). These 129 proteins comprise proteins on the

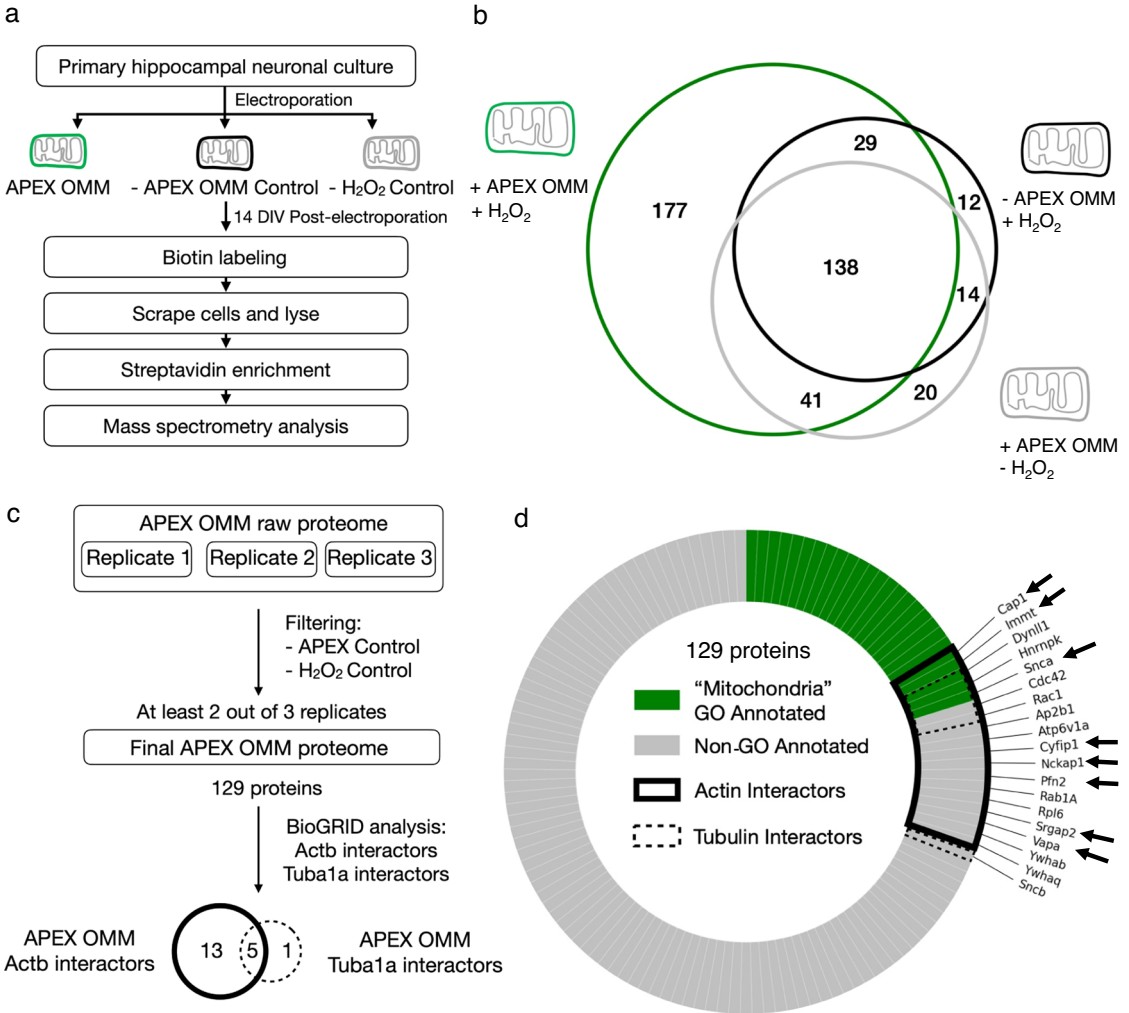

**Fig. 2 | Identifying actin-interacting proteins on the outer mitochondrial membrane. a** Experimental workflow for APEX-OMM proteome labeling in primary hippocampal neuronal cultures for mass spectrometry analysis. **b** Representative biological replicate (of **c**, **d**) showing the OMM proteome yield (green) after subtracting the proteins measured in Controls–in the absence of APEX-OMM (black) and the absence of hydrogen peroxide (gray). The proteins detected in Controls are endogenous biotinylated proteins (pyruvate carboxylase, 3-methylcrotonyl CoA carboxylase, propionyl CoA carboxylase, and acetyl CoA carboxylase) and non-specific proteins binding to streptavidin beads during enrichment. As we followed a stringent criterion of excluding the proteins found in Controls from experimental samples, irrespective of their intensities (see Methods), there might be an

overestimation of proteins in the two Control samples. **c** Flowchart of the mass spectrometry proteome analysis of three biological replicates, following filtering for Controls as in (**b**), yielding 129 proteins from two out of three biological replicates and identifying actin (Actb) and tubulin (Tuba1a) interactors on the OMM using BioGRID. **d** 129 proteins of the OMM proteome analyzed for gene ontology (GO) annotation for the term "Mitochondria" ("Mitochondria GO Annotated, green), and proteins not GO annotated as "Mitochondria" (Non-GO Annotated, gray). 18 proteins identified in BioGRID as actin interactors (Actin Interactors, black line), 6 proteins identified in BioGRID as tubulin interactors (Tubulin Interactors, dotted black line), and the 8 actin-interacting OMM proteins selected for the next round of screening (black arrows). n: 3 biological replicates, 3 animals.

mitochondrial membrane (26 proteins with 'Mitochondria' GO annotation, Fig. 2d, green, Supplementary Table 1) and proteins interacting with the mitochondrial membrane (103 proteins without 'Mitochondria' GO annotation, Fig. 2d, gray, Supplementary Table 1). Of these 129 proteins, 69 (53%) overlap with the OMM proteome of HEK cells[26], validating the sensitivity of our method for known OMM interacting proteins (Supplementary Table 1, see Discussion). Furthermore, only 22 proteins (17%) of our OMM proteome overlap with the most abundant soluble proteins in the neuropil[33], and only 31 proteins (24%) of our OMM proteome overlap with all soluble proteins in the neuropil[33], confirming that the OMM proteome is not proteins merely bumping into mitochondria by chance with no functional interaction and APEX-OMM labeling is not too promiscuous that it only results in soluble proteins (Supplementary Table 1).

We further analyzed the 129 proteins using a curated repository for protein interactions, BioGRID[34], to identify known actin and tubulin protein interactors (see Methods). This analysis resulted in 13 exclusive actin-interacting proteins, 1 exclusive tubulin-interacting protein, and 5 actin and tubulin interactors (Fig. 2c, d, Supplementary Tables 1, 2). From these proteins, we focused on the following 8 proteins—Cap1, Immt, Snca, Cyfip1, Nckap1, Pfn2, Srgap2, and Vapa, for further analysis (Fig. 2d, Supplementary Table 2, gray). Most proteins with multiple roles in basic cellular function and signaling, such as Rab1a, Rac1, Rpl6,

Ywhab, Ywhaq, etc., were not followed up for this study (Fig. 2d, Supplementary Table 2, non-gray).

## VAP is required for mitochondria-actin tethering in dendrites

We then determined the requirement of the 8 protein candidates in tethering mitochondria to actin in dendrites. We genetically knocked down each candidate protein and measured the mitochondrial-actin interaction efficiency in dendrites and axons. For this purpose, we established a method based on Lifeact-GFP[35], an actin-visualization marker. Lifeact-GFP has an actin-interacting domain and can be targeted to the OMM (Fis1-Lifeact-GFP)[28]. At reduced expression levels of Lifeact-GFP in the OMM, only the Lifeact-GFP that binds and is recruited near actin and is, therefore, immobile is detectable by fluorescence (Fig. 3a, b). In contrast, the low fraction Lifeact-GFP that freely diffuses on the OMM and is mobile is undetectable. To image Fis1-Lifeact-GFP, we employed high-resolution Airyscan confocal that revealed the fraction of the OMM area colocalizing with actin, which we measure as a proxy for mitochondria-actin interaction (Fig. 3a, b, see Methods). The expression of Fis1-Lifeact-GFP did not cause any abnormal actin aggregation, actin enrichment near mitochondria, or disruption of actin dynamics in dendrites (Supplementary Fig. 3a, b, see Methods). Furthermore, lengthening (using DRP1 KD[36–38]) or shortening (using MFF OE[6,39]) mitochondria or knocking down the

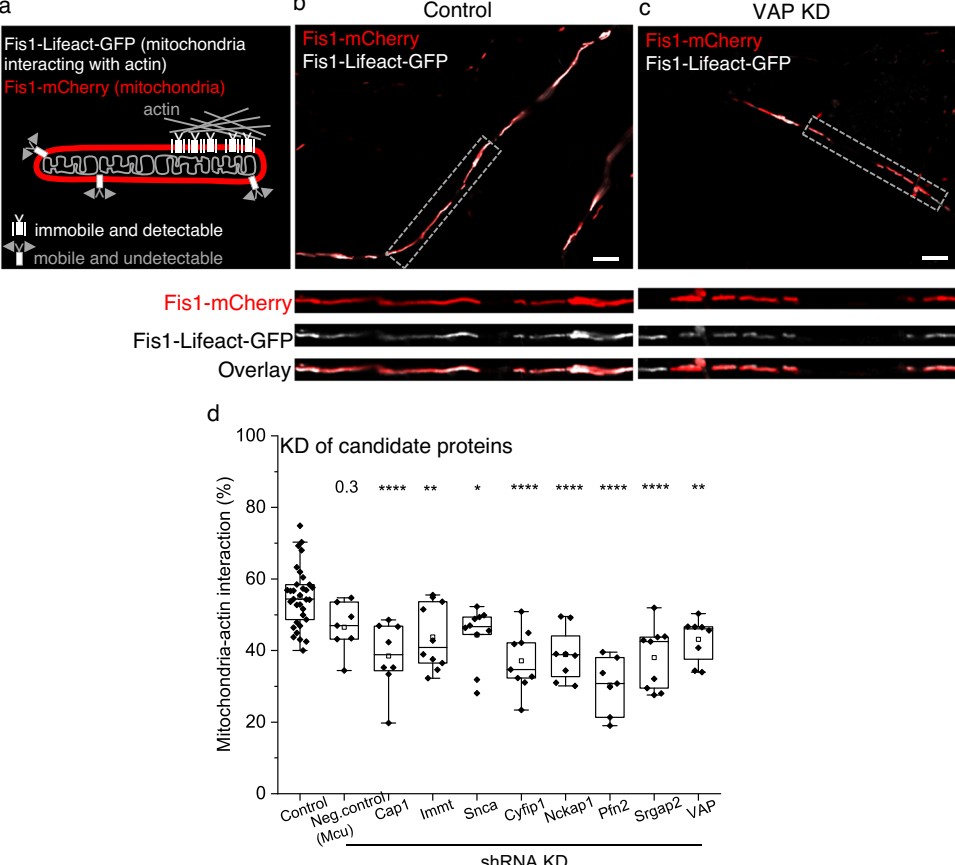

**Fig. 3 | Identifying proteins required for mitochondria-actin tethering in dendrites. a** Cartoon illustrating the labeling strategy for visualizing mitochondria and mitochondria-actin interaction regions. Representative Airyscan confocal image (of **d**) of a neuronal dendrite transfected with Fis1-mCherry (red), Fis1-Lifeact-GFP (white), and Control shRNA (**b**, Control) or VAP shRNA (**c**, VAP KD) showing fewer mitochondrial regions interacting with actin (white) in VAP KD (**c**) compared to Control (**b**). The gray dashed box depicts the straightened dendritic segment magnified for better visualization (inset). Scale bar, 5 μm. **d** Significant reduction in the average mitochondria-actin interaction percentage in all 8 protein knockdown conditions compared to Control shRNA-expressing dendrites. The mitochondrial uniporter (Mcu) was used as a negative Control, given its localization in the mitochondrial inner membrane and the absence of any known actin association. *n* in dendrites, animals: 33, 18 (Control), 7, 2 (Neg. control), 8, 3 (Cap1), 10, 4 (Immt), 10, 6 (Snca), 9, 1 (Cyfip1), 8, 2 (Nckap1), 7, 2 (Pfn2), 9, 6 (Srgap2), 8, 5 (VAP). One-way ANOVA, Tukey test, *p*-values: 0.31 (Neg. control), $8.75 \times 10^{-5}$ (Cap1), 0.01 (Immt), 0.02 (Snca), $4.71 \times 10^{-6}$ (Cyfip1), $1.36 \times 10^{-4}$ (Nckap1), $2.97 \times 10^{-8}$ (Pfn2), $1.65 \times 10^{-5}$ (Srgap2), and 0.01 (VAP). Source Data files are provided.

actin proteins required for mitochondrial fission (Inf2 KD[40] or Spire 1 KD[41]) did not affect mitochondria-actin interaction (Supplementary Fig. 3c). These data suggest this measurement is insensitive to mitochondrial size and fission status.

As a positive control, neurons were treated with actin depolymerizing agents, Cytochalasin (Cyto-D) and Latrunculin-B (Lat-B), to confirm the ability of Fis1-Lifeact-GFP to detect a reduction in mitochondria-actin interaction (Supplementary Fig. 3d). As a negative control for the Fis1-Lifeact-GFP measurements, neurons were treated with a microtubule depolymerizing agent that does not affect actin, Nocodazole (Noco). Noco treatment did not affect mitochondria-actin interaction (Supplementary Fig. 3d).

Knocking down the 8 protein candidates, Cap1, Immt, Snca, Cyfip1, Nckap1, Pfn2, Srgap2, and VAP reduced the mitochondria-actin interaction in dendrites (Fig. 3b–d), suggesting their significance in tethering mitochondria to actin. Vap is expressed as two paralogs, Vapa and Vapb[29]. As one paralog can compensate for the absence of the other[37], we deleted both paralogs (denoted as VAP in the text unless specified otherwise) for our gene deletion experiments. VAP KD or KO did not disrupt actin in dendrites, spines, and axons, nor actin dynamics in spines (Supplementary Fig. 3e–g, see Methods).

Axonal mitochondrial interaction with actin was unaffected when treated with actin depolymerizing agents and when the 8 protein candidates were knocked down (Supplementary Fig. 3h, i). This data suggests that the identified 8 protein candidates are exclusive to mitochondrial-actin tethering in dendrites, not axons. These data are consistent with our previous observations that axonal mitochondria stabilize on cytoskeletal depolymerization while dendritic mitochondria destabilize[6,9], suggesting that different mechanisms tether mitochondria in axons compared to dendrites (see Discussion). In addition, VAP KD did not affect mitochondria-actin interaction in neuronal soma (Supplementary Fig. 3j, see Methods), suggesting that the VAP-dependent mitochondria-actin tethering is exclusive to dendritic compartments, not soma.

## VAP stabilizes mitochondrial compartments in dendrites

The 8 proteins tethering mitochondria to actin might not necessarily stabilize mitochondrial compartments in dendrites. Therefore, we further refined the 8 protein-candidate list by identifying the protein(s) required for stabilizing mitochondria. To measure mitochondrial stabilization, we used our previously established photoactivation method[6] (Fig. 4a, b). Briefly, mitochondria were photoactivated in dendrites of neurons expressing photoactivatable GFP targeted to the mitochondrial matrix (Mito-PAGFP) and monitored for 60 min (Fig. 4a, b). The mitochondrial compartment length was measured as the spatial spread of the photoactivated fluorescent signal (Fig. 4a, b, Supplementary Fig. 4a, b). Mitochondrial compartment stability was measured by the photoactivated compartment's fluorescence intensity over 60 min (Fig. 4a–d, Supplementary Fig. 4d, see Methods).

We knocked down 3 of the 8 protein candidates (VAP, Snca, Srgap2) for this study and measured mitochondrial compartment size and stability in dendrites. Knocking down VAP, Snca, and Srgap2 shortened the length of mitochondrial compartments and mitochondria (Fig. 4a, b, Supplementary Fig. 4a–c, see Methods). We then measured mitochondrial stability in VAP KD (and KO), Snca KD, and Srgap2 KD dendrites by monitoring the photoactivated compartment fluorescence over time and measuring the compartment stability index (Fig. 4b–d, Supplementary Fig. 4d–h, see Methods). Mitochondria in Control dendrites showed a modest decrease in photoactivated compartment fluorescence 60 minutes postphotoactivation, perhaps corresponding to fluorescent protein leak from mitochondria during basal-level mitochondrial dynamics such as fission and fusion and not due to photobleaching (Fig. 4c). Meanwhile, dendrites lacking VAP showed a dramatic decrease in photoactivated compartment fluorescence 60 minutes postphotoactivation, corresponding to destabilized

mitochondrial compartments (Fig. 4a–d, Supplementary Fig. 4d). On the other hand, knocking down Snca and Srgap2 did not destabilize mitochondrial compartments (Fig. 4a–d). In addition, the mitochondrial compartment length showed a decreasing trend when comparing the 5- and 60-min postphotoactivation periods in VAP KD but not in SNCA and SRGAP2 KD (Supplementary Fig. 4a, b). These results are similar to the mitochondrial compartment length shortening and destabilization observed on actin perturbation[6], suggesting VAP's role in actin-tethering and mitochondrial stabilization in dendrites.

## Vapb is enriched near mitochondria in dendrites but not in axons

VAP is an endoplasmic reticulum (ER)-associated protein that tethers ER with other organelles such as endosomes and mitochondria, and its dysfunction is implicated in Amyotrophic Lateral Sclerosis (ALS), Parkinson's Disease (PD), and frontotemporal dementia[29–31,42,43]. As VAP is expressed as two paralogs, Vapa and Vapb, and we deleted both for our gene deletion experiments, the local association of each paralog with mitochondria is unclear in dendritic and axonal compartments.

We, therefore, visualized endogenous Vapa and Vapb by immunostaining and measured its association with mitochondria in both dendrites and axons in fixed neurons. Vapa (A, immunodetection) is enriched near dendritic and axonal mitochondria (Fig. 5a–c, see below, see Methods). However, Vapb (B, immunodetection) is only enriched near dendritic mitochondria and not axonal mitochondria (Fig. 5d–i). As Vapa is equally enriched near the long dendritic and short axonal mitochondria, the enrichment of Vapb only near dendritic mitochondria is not due to the longer length of dendritic mitochondria compared to axonal mitochondria. Visualization of Vapb in live neurons (B_{live}, Vapb-emerald, or Vapb-HaloTag-635) also confirmed its enrichment near dendritic mitochondria, including moving mitochondria following fission, but not near axonal mitochondria (Fig. 5i, Supplementary Fig. 5a–e). We also confirmed that Vapb is not distributed throughout the ER and is enriched near mitochondria (Supplementary Fig. 5f, g). The enrichment of Vapb only near dendritic mitochondria and its overall lower expression compared to Vapa[33,44] could be why we did not detect it in our APEX-OMM proteome (see Discussion). Our results suggest that Vapb, exclusively enriched near dendritic mitochondria, might determine the dendrite-specific role of VAP in stabilizing mitochondria.

## VAP determines the dendritic segment supported by mitochondria

Next, we determined if VAP-dependent mitochondrial stabilization can locally support synaptic plasticity. Mitochondrial compartments are stable for ~60 min and are essential to fuel protein synthesis-dependent synaptic plasticity within 30 μm dendritic segments[6]. So, we hypothesized that if VAP is required for the temporal and spatial stabilization of dendritic mitochondria, the absence of VAP will affect both the temporal sustenance of synaptic plasticity and the 30 μm dendritic segment exhibiting synaptic plasticity.

We induced spine plasticity using two-photon glutamate uncaging (0.5 Hz,120 pulses) in PSD95- or Homer2- positive spines in the presence of forskolin to activate the PKA pathway to induce long-term structural plasticity[19] (Fig. 6a). We then measured spine size increase in the plasticity-induced spine and adjacent, uninduced spines in the same dendritic segment (Fig. 6a–g). We also measured spine size from uninduced spines in sister dendrites as a Control (Fig. 6c).

The absence of VAP reduced spine morphological plasticity in the plasticity-induced spine, indicating the importance of stable mitochondrial compartments at the base of the plasticity-induced spine (Fig. 6a–g). Interestingly, VAP deletion did not affect the early stages of synaptic plasticity induction as the measured spine-head width increase was similar to that of Control spines at $t = 2$ and 12 min (Fig. 6c–f). However, post-plasticity induction over longer durations of

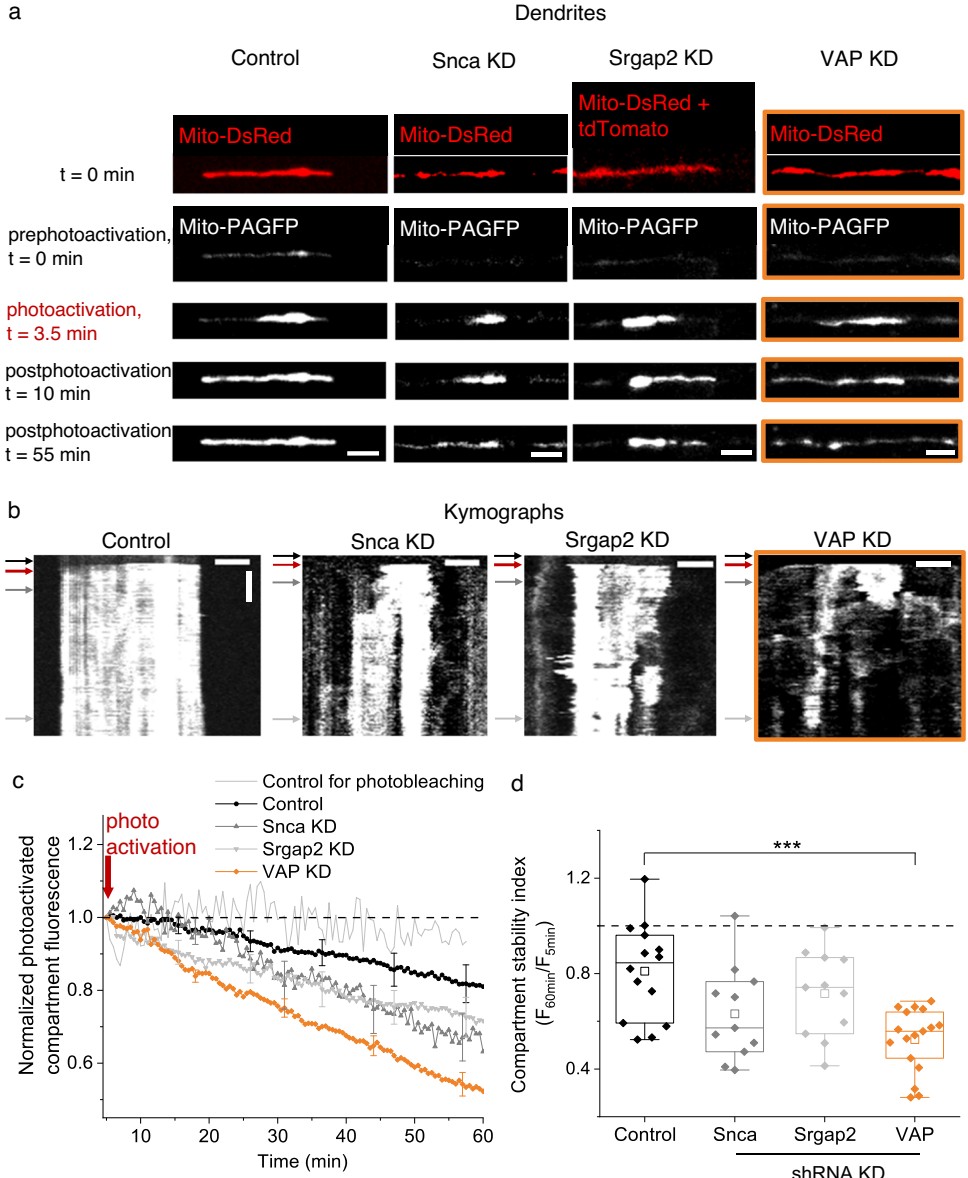

**Fig. 4 | VAP spatially stabilizes mitochondria in dendrites. a** Representative images (of **c**, **d**) of straightened dendrites expressing Mito-DsRed (red), Mito-PAGFP (white) and Control shRNA (Control), Snca shRNA (Snca KD), Srgap2 shRNA (Srgap2 KD) or VAP shRNA (VAP KD, orange box) before (prephotoactivation, *t* = 0 min), at (photoactivation, *t* = 3.5 min), and after photoactivation (post-photoactivation *t* = 10 min, 55 min). The Mito-DsRed signal was used to locate mitochondria for photoactivation at *t* = 0 min. In Srgap2 KD dendrites, the Mito-DsRed signal is contaminated by the cytosolic tdTomato signal coexpressed by the Srgap2 shRNA plasmid. Scale bar, 5 μm. **b** Representative kymographs (of **c**, **d**) of the photoactivated mitochondrial compartments in (**a**) show stable mitochondrial compartments in Control; shortened and modest destabilization of mitochondrial compartments in Snca and Srgap2 KD; and shortened and destabilized mitochondrial compartments in VAP KD (orange box). The black arrow denotes the pre-photoactivation, the red arrow denotes the photoactivation, and the dark and light gray arrows denote the postphotoactivation time points. Horizontal scale bar, 5 μm. Vertical scale bar, 10 min. **c** The average time course of photoactivated mitochondrial compartment fluorescence shows a modest decrease in Control (black), a moderate decrease that was not statistically significant in Snca (gray) and Srgap2 KD (light gray), and a dramatic and statistically significant decrease in VAP KD (orange). n in dendrites, animals: 14, 3 (Control), 11, 2 (Snca KD), 11, 2 (Srgap2 KD), 17, 4 (VAP KD). While reduction in photoactivated compartment fluorescence could be due to continuity between mitochondria due to fusion, this is minimal in Control; and in VAP KD, the mitochondria are shorter, perhaps due to decreased fusion (see Discussion). Control for photobleaching (light gray without symbol, n: 4 regions, 3 animals) was measured from static, photoactivated mitochondrial regions, showed a negligible decrease, suggesting minimal photobleaching during imaging. The red arrow denotes the photoactivation time point. **d** The mitochondrial compartment stability index showed a statistically significant decrease in VAP KD dendrites compared to Control, Snca KD, and Srgap2 KD. n: same as in (**c**). One-way ANOVA, Tukey test, *p*-value: $2.3 \times 10^{-4}$. Source Data files are provided.

*t* = 22 to 62 min, spines lacking VAP were unable to sustain the structural plasticity (Fig. 6b–e, g).

VAP depletion did not affect spine density and showed a significant but slight increase in spine size at baseline compared to Control neurons (Supplementary Fig. 6a, b). The structural plasticity defect in VAP KO spines is not due to the slight increase in baseline spine size in VAP KO compared to Control, as this increase is small, in the range of spine size fluctuations observed in Control, uninduced spines (Fig. 6c, Supplementary Fig. 6b–e). Nor is the structural plasticity defect in VAP KO due to any deficiencies in baseline spine calcium, spine calcium influx, or ER calcium release at the base of the plasticity-induced spine (Supplementary Fig. 6f–i). However, mitochondrial calcium influx at the base of the plasticity-induced spine was significantly affected in VAP KO compared to Control,

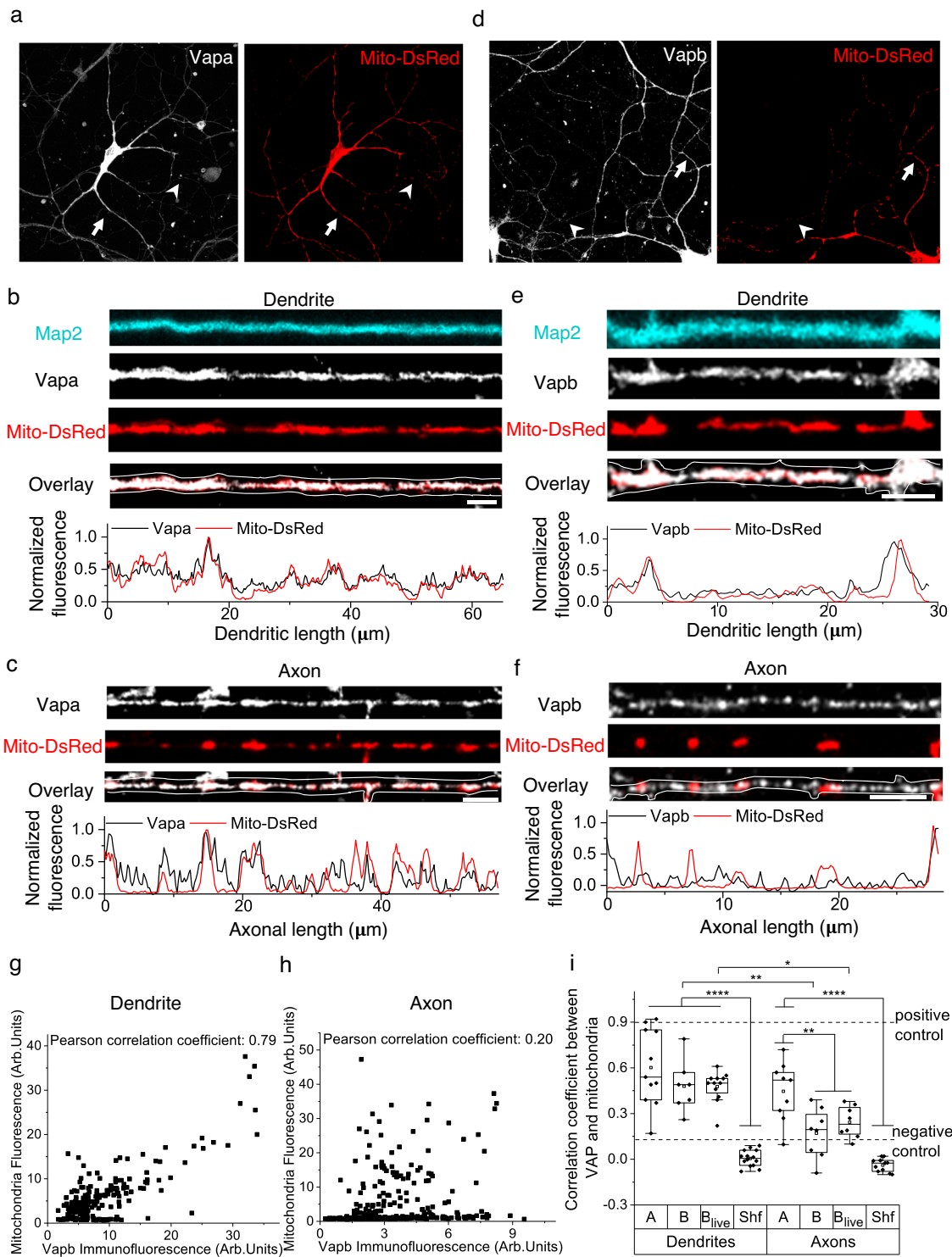

suggesting that mitochondrial energy production might be affected (Supplementary Fig. 6j, see Discussion). Our earlier work showed that depletion of a local 30 μm-mitochondrial compartment in a dendritic segment affected both synaptic plasticity formation and sustenance[6]. In contrast, here, we only manipulated the VAP-dependent stabilization of mitochondrial compartments, and it does not affect the early stages of synaptic plasticity formation but only its long-term sustenance. These data suggest the importance of stabilized mitochondrial compartments and their local activity in supporting the long durations of synaptic plasticity formation and maintenance.

Furthermore, to determine the spatial dendritic segment supported by VAP-dependent mitochondrial stabilization, we investigated the ability of the uninduced spines adjacent to the plasticity-induced spine to exhibit structural plasticity, a phenomenon otherwise known as clustered synaptic plasticity. In Control neurons, spines adjacent to the plasticity-induced spine within 0-15 μm showed structural plasticity consistent with earlier observations[19–24], with a similar trend in spines within 15–30 μm distance and no apparent structural plasticity in spines within 30–45 μm distance, at $t = 2$ and 62 min post-plasticity induction (Fig. 6d–g). However, on VAP deletion, the ability of the

**Fig. 5 | Vapb is enriched near mitochondria in dendrites but not in axons.**
Representative images (of **i**) of neurons immunostained for Vapa (white) and mitochondria (Mito-DsRed, red) (**a**); and Vapb (white) and mitochondria (Mito-DsRed, red) (**d**). Arrows depicting dendrites and arrowheads depicting axons, straightened and magnified for better visualization in (**b**–**f**). Representative line profiles (of **i**) showing local enrichment of Vapa near dendritic (**b**, arrow in **a**) and axonal mitochondria (**c**, arrowhead in **a**) and Vapb near dendritic (**e**, arrow in **d**) but not near axonal mitochondria (**f**, arrowhead in **d**). Map2 (cyan) was used to trace the dendrite, and the Vapa and Vapb signal was used to trace the axon (white lines). Vapa and Vapb antibodies were validated in Vapa and Vapb knockdown neurons, respectively. Scale bar, 5 μm. Representative correlation (of **i**) between individual fluorescent pixel intensities of mitochondria (Mito-DsRed) and Vapb show a strong correlation in the dendrite (**g**) but not in the axon (**h**). **i** The average correlation coefficient measured shows a strong correlation between mitochondria and Vapa

(A), Vapb (B) in fixed dendrites, and Vapb in live dendrites ($B_{live}$, Vapb-emerald), compared to shuffled control (Shf, see Methods). In axons, a strong correlation between mitochondria and A was observed, but not between mitochondria and B and $B_{live}$ compared to Shf. Furthermore, the correlation coefficient between Vapb (B and $B_{live}$) and mitochondria in dendrites is statistically significant compared to that in axons. n in dendrites, animals: 11, 2 (A), 7, 3 (B), 12, 4 ($B_{live}$), 14, 4 (Shf). n in axons, animals: 9, 2 (A), 8, 2 (B), 5, 1 ($B_{live}$), 11, 3 (Shf). One-way ANOVA, Tukey test, p-values: 0.00261 (B den versus ax), 0.01644 ($B_{live}$ den versus ax), <0.0001 (A, B, $B_{live}$ den versus Shf den), <0.0001 (A ax vs Shf ax), 0.00338 (A ax vs B and $B_{live}$ ax). As a positive control, the correlation between two different fluorescent tags targeted to the mitochondrial matrix (Mito-DsRed and Mito-PAGFP) was used, which provided the upper limit of the correlation coefficient. As a negative control, the correlation between a mitochondrial (Mito-DsRed) and a non-mitochondrial tag was used for the lower limit of the correlation coefficient. Source Data files are provided.

adjacent spines to exhibit structural plasticity was affected as early as $t = 2$ minutes post-plasticity induction (Fig. 6d–g). The structural plasticity defects in adjacent spines in VAP KO are not due to baseline spine size differences between VAP KO and Control spines (Supplementary Fig. 6d, e). These data suggest that mitochondrial compartment length shortening and destabilization and reduced mitochondrial density (Fig. 4a–d, Supplementary Figs. 4a–d, 6k) in the absence of VAP affect the long-term plasticity formation in stimulated and adjacent spines within a 30 μm distance. In summary, these results indicate that VAP functions as a spatial stabilizer that temporally sustains synaptic plasticity for up to ~60 min and as a spatial ruler that determines the ~30 μm dendritic segment supported by mitochondria during synaptic plasticity (Fig. 7).

## Discussion

Mitochondrial dysfunction is associated with various disease states. As distal synapses rely on local energy supplies such as mitochondria, even brief interruptions in their local availability can result in long-lasting changes in synaptic and network properties, as observed in neurodegenerative diseases, including Alzheimer's (AD), PD, and ALS[45–47]. Using recent advances in subcellular proteomic labeling, high-resolution imaging techniques to quantify mitochondrial-actin interactions, and stimulation of individual spines using two-photon glutamate uncaging for measurements of synaptic plasticity, we probe various mechanisms determining the spatial organization of dendritic mitochondria and its role in supporting synaptic plasticity. We demonstrate the following: (1) identification of mitochondrial-actin protein interactors in neurons; (2) identification of mitochondrial-actin tethering proteins that are exclusive to neuronal dendrites; (3) distinct role of VAP in spatially stabilizing mitochondria in dendrites; (4) Vapb's exclusive enrichment near dendritic mitochondria, and not near axonal mitochondria; (5) disruption of sustained spine morphological plasticity in plasticity-induced spines and adjacent spines within 30 μm upon VAP gene deletion; (6) VAP serves as the spatial stabilizer that determines the temporal sustenance of synaptic plasticity for 60 min and a spatial ruler that determines the 30 μm dendritic segment supported by mitochondria during synaptic plasticity.

Using APEX-based proximity labeling, we identified 129 proteins in the neuronal OMM proteome, including proteins present on the OMM and interacting with the OMM (Figs. 1, 2). Of these 129 proteins, 69 proteins (53%) overlap with the OMM proteome of HEK cells[26], which includes 7 of the 8 mitochondrial-actin interacting proteins we investigated in our experiments (Cap1, Snca, Cyfip1, Nckap1, Pfn2, Srgap2, VAP, except for Immt) (Supplementary Table 1). The remaining proteins we identified in the OMM proteome might be neuron-specific. The proteins that were found in the HEK cell OMM proteome but not in our proteome might either be HEK cell-specific proteins, proteins undetected due to different mass spectrometry analysis strategies, or low expressing proteins that could not be detected in neurons (neurons being non-dividing cells generally provide lower proteome yield

compared to HEK cells). Furthermore, the BioGRID database we used to filter for actin-interacting proteins is constantly updated, and not all actin interactors are fully annotated or identified. Our mitochondrial-actin interactor list will, therefore, be updated with the growing literature in the future.

Dendritic mitochondria differ from axonal mitochondria in size, shape, and motility. Dendritic mitochondria can be as long as 30 μm in length, where multiple mitochondrial filaments stack against each other to make one physical compartment (Figs. 3b, 4a, 5b, e, 6d, Supplementary Figs. 1b, 3a, e, 4a–c, 5a)[6,16,17]. Dendritic mitochondria are also spatially stable for 60 to 120 min and during development (Fig. 4a–d, Supplementary Fig. 4a, b, d)[6,48]. On the other hand, axonal mitochondria are short, between 0.5 and 2 μm in length, they are not stacked, and are motile within the same duration of 60 to 120 min (Fig. 5c, f, Supplementary Figs. 3f, 5b, c)[6,12,16,18]. However, it has been shown that mitochondria reach stable positions during axonal maturation and branching[10,14,49]. In addition, the mechanisms by which dendritic and axonal mitochondria interact with the cytoskeleton are also strikingly different. When the cytoskeleton (actin or microtubule) is perturbed, stable dendritic mitochondria get destabilized, but in contrast, the motile axonal mitochondria get stabilized[6,9]. This observation suggests that dendritic mitochondria interact differently with the cytoskeleton than axonal mitochondria.

Interestingly, the 8 proteins we characterized as essential for mitochondrial-actin tethering in dendrites are not required for mitochondrial-actin tethering in axons. This result suggests that mitochondria have different tethering and stabilization rules in axons compared to dendrites. For example, syntaphilin, which facilitates mitochondrial docking via the microtubule, is an axon-specific protein not present in dendrites[11]. Similarly, FHL2 was recently shown to anchor mitochondria to actin when switched from low glucose to high glucose concentrations in axons[13]. Since the APEX-OMM proteome labeling was done at uniform glucose concentrations without any induced glucose influx, and due to the lower concentrations of FHL2 in the neuropil proteome[44], we do not see it in our OMM proteome list. However, while we did not investigate FHL2 in dendrites in this study, knocking down FHL2 in neurons also affects mitochondrial-actin interaction in dendrites (unpublished data), emphasizing that it may play a dual role in both neuronal compartments. Future studies should, therefore, identify more proteins that play a specific or unifying role in mitochondrial cytoskeletal tethering and stabilization in dendrites and axons. Furthermore, the current study only focused on mitochondrial-actin tethering and stabilization proteins in dendrites. The mitochondrial-tubulin interacting proteins and their role in dendritic mitochondrial tethering and stabilization should be investigated in future studies.

VAP is an ER resident protein found in membrane contact sites with various organelles such as Golgi, mitochondria, peroxisomes, endolysosomes, and plasma membrane. It plays an essential role in lipid transfer between these organelles[29]. This role of

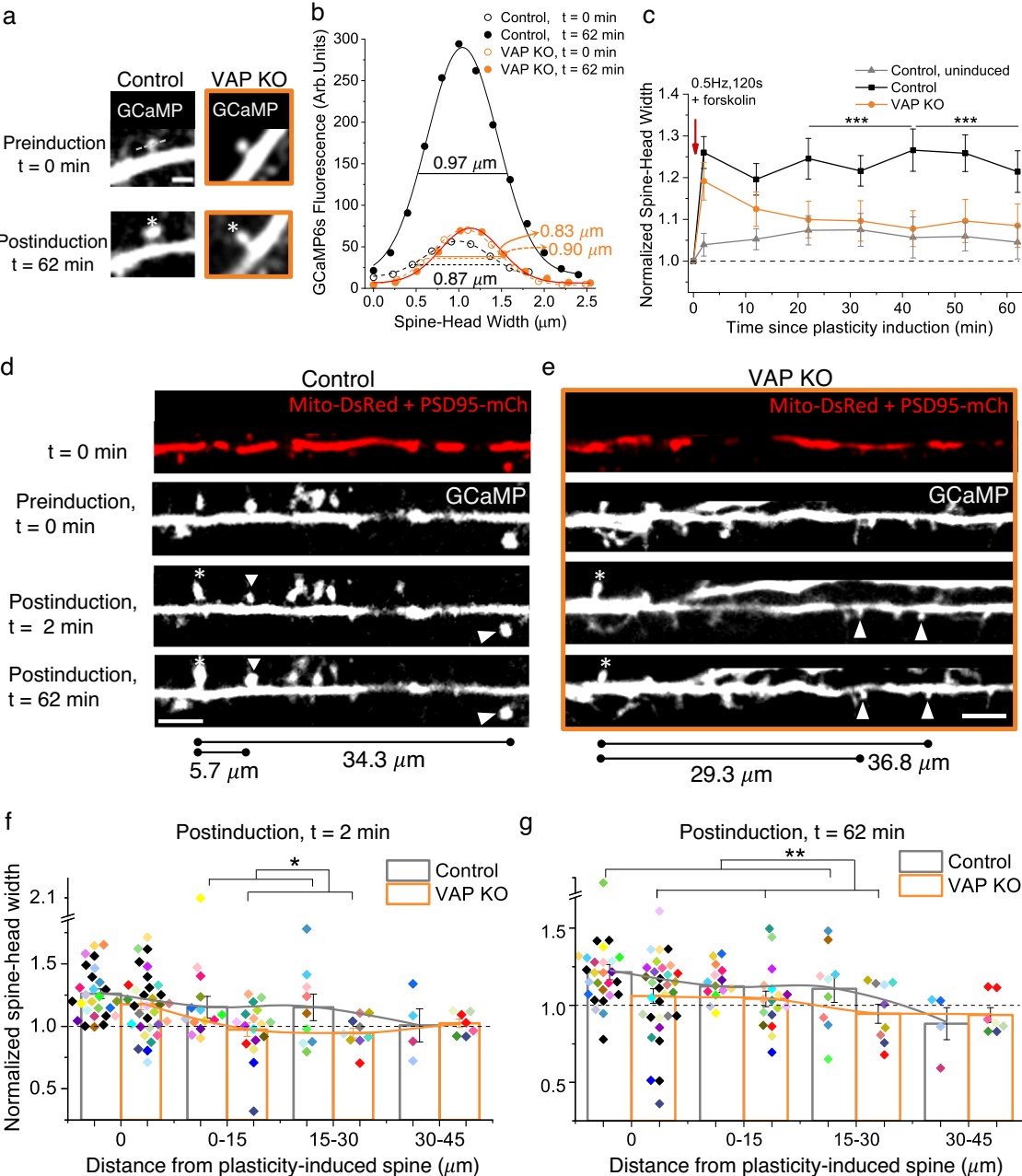

**Fig. 6 | VAP's absence affects synaptic and clustered synaptic plasticity.**
**a** Representative plasticity-induced spine (white asterisk) (of **c**, **f**, **g**) measured at 0 and 62 min showed an increase in spine-head width in Control but not in VAP KO. Scale bar, 2 μm. **b** GCaMP fluorescence along the line crossing the spine-head center (gray dashed line in **a**) to measure spine-head width in Control (black) and VAP KO spines (orange), before (empty circles, $t = 0$ min) and after plasticity induction (full circles, $t = 62$ min). **c** The average time course showed an increase in spine-head width in Control (black) but not in VAP KO (orange) at 62 min post-plasticity induction and in Control, uninduced spines (gray). n in spines, animals: 27, 11 (Control), 33, 7 (VAP KO), 19, 5 (Control, uninduced). One-way ANOVA, Tukey test, *p*-values: $8.77 \times 10^{-4}$ (22-42 min) and $3.36 \times 10^{-4}$ (42–62 min). Paired sample *t*-test between $t = 2$ and 62 min in VAP KO, *p* values: 0.06 (normalized data), 0.04 (unnormalized). **d**, **e** Representative plasticity-induced spines (white asterisk) (of **f**, **g**) measured at 0, 2, and 62 min showed an increase in spine-head width in plasticity-induced and adjacent, PSD95- or Homer2-positive, uninduced spines (white arrowhead) within 30 μm in Control neurons. VAP KO neurons exhibited structural plasticity in the plasticity-induced spine (white asterisk) 2 min post-induction but not in adjacent, uninduced spines (white arrowhead) and 62 min postinduction. Scale bar, 5 μm. **f**, **g** Histograms showing an increase in spine-head width 2 and 62 min post-plasticity induction, within 30 μm from plasticity-induced spines, in Control (gray) and 2 min post-plasticity induction in VAP KO (orange) (**f**), but not in adjacent spines and 62 min post-plasticity induction in VAP KO (orange) (**g**). Individual points represent spines and are color-coded by neurons. For Control, n in spines, animals: 27, 11 (0 μm), 14, 8 (0–15 μm); 9, 5 (15–30 μm); 4, 2 (30–45 μm). For VAP KO, n in spines, animals: 32, 7 (0 μm); 17, 4 (0–15 μm); 9, 5 (15–30 μm); 7, 4 (30–45 μm). One-way ANOVA, Tukey test, *p*-values: 0.031 (**f**), 0.003 (**g**). Source Data files are provided.

VAP could explain the shortened mitochondria in VAP KD and KO (Figs. 3c, 4a, b, 6e, Supplementary Figs. 3e, 4a–c), as lipid transfer between ER and mitochondria is required for mitochondrial fusion[50]. VAP is identified as an actin interactor in the BioGRID database[34] based on human interactome network analysis studies[51,52]. However, how VAP tethers actin to mitochondria is unclear. It has been shown that VAP-dependent actin nucleation is required for ER-endosome contacts[42]. So, a similar actin nucleation process might

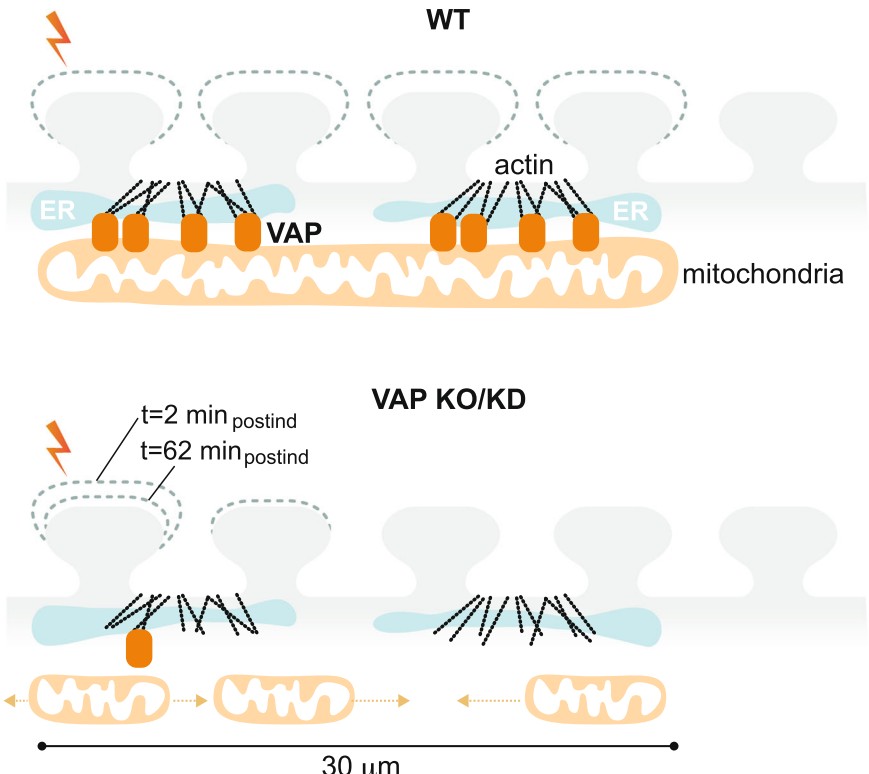

**Fig. 7 | VAP spatially stabilizes mitochondria to locally support synaptic plasticity.** Illustration showing the significance of VAP (orange rounded rectangle) as a spatial organizer in stabilizing mitochondrial compartments (light orange) to support plasticity-induced spines (lightning bolt) and adjacent spines (light gray). The WT dendritic segment with normal VAP expression and mitochondrial stabilization exhibits spine structural plasticity in plasticity-induced and adjacent spines within 30 μm. VAP deletion shortens and destabilizes mitochondria (short light orange mitochondria that are mobile with light orange dashed arrows) in the dendrites and abolishes spine structural plasticity sustenance in both plasticity-induced and adjacent spines within 30 μm. VAP is a spatial organizer in stabilizing mitochondrial compartments via ER-actin-tethering (cyan ER and black actin filaments) for sustained synaptic plasticity formation. VAP also functions as the spatial ruler, determining the 30 μm dendritic segment supported by a mitochondrial compartment. These data emphasize the importance of VAP as a molecular tether in locally stabilizing a mitochondrial compartment to support the synaptic and clustered synaptic plasticity required for learning and development.

facilitate ER-mitochondria contacts and needs to be tested in future studies.

VAP is expressed as two paralogs, Vapa and Vapb. Vapa is generally expressed in high levels compared to Vapb, so Vapa can compensate for the absence of Vapb[29]. So, we knocked down both Vapa and Vapb in our photoactivation and structural plasticity measurements. However, immunolocalization studies of the individual Vapa and Vapb paralogs reveal that Vapb is exclusively enriched near dendritic mitochondria. In contrast, Vapa is enriched both near dendritic and axonal mitochondria (Fig. 5, Supplementary Fig. 5a–e). This result suggests that Vapb might be the dendrite-specific paralog specializing the tethering and stabilization rules of mitochondria in dendrites. However, we did not find Vapb in our APEX-OMM proteome dataset, possibly due to its low expression levels compared to Vapa and its specific enrichment only near dendritic mitochondria (Fig. 5, Supplementary Fig. 5, Supplementary Table 1). Indeed, recent transcriptome and translatome studies confirm that Vapb is lowly expressed compared to Vapa in hippocampal neurons[33,44].

ER-Mitochondria contact sites (ER-MCS) are crucial communication hotspots for ER-mitochondria calcium signaling, mitochondrial ATP production, and mitochondrial fission[53–55]. Vapb is a well-known ER-MCS tether, where the ER protein Vapb interacts with the protein tyrosine phosphatase interacting protein 51 (Ptpip51) on the mitochondrial outer membrane[56]. Consistent with previous results[56], we find that VAP's absence affects mitochondrial calcium influx, while ER calcium release is unaffected (Supplementary Fig. 6i, j). As mitochondrial calcium is critical in activating enzymes of the Krebs cycle for ATP generation[57,58], we hypothesize that VAP's absence affects local mitochondrial ATP generation due to reduced mitochondrial calcium influx. In addition, the shortened and destabilized mitochondria and reduced mitochondrial density in VAP's absence might affect the sustained ATP required to support synaptic plasticity for long hours (Figs. 4a–d, 6a–g, Supplementary Figs. 4a–d, 6c, e, k). Future studies investigating mitochondrial and spine ATP measurements in VAP KO neurons during synaptic plasticity should address these questions.

Super-resolution imaging of ER-MCS contacts in non-neuronal cells shows that nutrient deprivation reduces Vapb mobility at ER-MCS, potentially facilitating efficient trans-organelle metabolite transfer for energy production[59]. We could, therefore, hypothesize that in neuronal dendrites at the base of the plasticity-induced spine, where energy demands are high, reduced VAP mobility might stabilize ER-actin-mitochondria interactions for long periods for enhanced local energy production. Future studies investigating VAP dynamics at super-resolution[59] at ER-MCS and mitochondria-actin interaction sites during synaptic plasticity will shed light on these questions. Moreover, we find 18 proteins in our OMM proteome that are GO annotated as "Endoplasmic Reticulum" (Supplementary Table 1). These proteins, including the 3 proteins, GO annotated as both "Mitochondria" and "Endoplasmic Reticulum" (Supplementary Table 1), should be further examined in the future for their role in ER-MCS and mitochondrial stabilization in neurons.

A P56S mutation in Vapb disrupts the ER-mitochondria interaction at ER-MCS and is implicated in ALS[56,60]. Most Vapb-related studies in ALS have focused on axons. Our findings on Vapb in dendrites, where they stabilize mitochondria to locally support and sustain synaptic plasticity, highlight its critical role in learning and memory. It brings a

new perspective to the motor learning impairments observed in ALS[60]. Recently, Vapb was also identified as a PD risk gene[61]. Future studies focusing on these mutations and their effect on mitochondrial stabilization, synaptic plasticity, learning, and memory will better understand the disease progression in ALS and PD.

Mitochondria represent a hugely underexplored organellar system in neurons. Notably, in dendrites and during synaptic plasticity, the role of mitochondria has been largely understudied. Using recent advances in proteomics and imaging, we have discovered a distinct function of VAP in spatially stabilizing mitochondria and its importance in plasticity formation in stimulated and adjacent spines. These findings raise a new set of questions on the spatiotemporal regulation of local energy supplies and how they adapt to the local energetic needs of a dendritic segment. Understanding these specialized mechanisms also adds new perspectives to the compartment-specific rules of dendrites that are important for dendritic computation, learning, and memory formation[62].

## Methods

### Animals
All experiments were performed according to the Max Planck Florida Institute for Neuroscience IACUC regulations (protocol number 22-005).

### Plasmid constructs
APEX-Matrix (APEX2 targeted to peptide sequence derived from COX4) and APEX-OMM (APEX2 targeted to MAVS peptide sequence) plasmids were obtained from the Ting Lab. DsRed-Mito (Clontech) was purchased, and then DsRed was replaced with EGFP to make the Mito-EGFP plasmid. The Fis1-Lifeact-GFP, Fis1-mCherry, and Spire1 KD plasmids were obtained from the Manor Lab, Mff-OE and Srgap2 KD plasmids from the Polleux Lab, VAP KD and Control plasmids from the Hoogenraad Lab, VAP KO and Control sgRNA plasmids from the Hoppa Lab, Vapb-emerald and Vapb-HaloTag plasmids from the Lippincott-Schwartz Lab, Inf2 KD plasmid from the Higgs Lab, ER-VSVG-GFP and PSD95-mCherry plasmids from Hanus C, and Homer2-mOrange plasmid from the de Juan-Sanz Lab. Lifeact-mCherry plasmid (gift from Michael Davidson), Mito-BFP plasmid, Drp1 KD plasmid (gift from David Kashatus), Snca KD and Control plasmids, Mito-PAGFP plasmid, and GCaMP6s plasmid were purchased from Addgene (plasmids 54491, 49151, 99385, 75437, 75438, 23348, 40753, respectively). All other shRNAs against *CAP1*, *IMMT*, *CYFIP1*, *NCKAP1*, and *PFN2* were purchased from Abbexa.

### Cell preparation, transfection, electroporation
Unless specified otherwise, all reagents were purchased from Sigma, and all stock solutions were stored at −20 °C. Sprague-Dawley rats (postnatal P0) were obtained from the in-house animal core facility of the Max Planck Florida Institute for Neuroscience. The sex of pups from which neurons were isolated was not determined as in previous publications in the field. Hippocampal regions were dissected and dissociated with the Worthington Papain Dissociation Kit (Worthington Biochemical Corporation, stored at 4 °C) with a modified manufacturer's protocol. Briefly, the digestion is done once for 30 min. Following trituration, the cells are centrifuged at $300 \times g$ for 5 min, resuspended in 2.4 ml of resuspension buffer (1.44 ml EBSS, 160 µl inhibitor, and 80 µl DNase) and 3.2 ml gradient, and then centrifuged at $53 \times g$ for 12 min. Neurons were plated on poly-D-lysine coated coverslips mounted on MatTek dishes at 60,000-80,000 cells/cm² density. Cultures were maintained in NGM (500 ml Neurobasal-A Medium, 10 ml B-27, and 5 ml Glutamax, from Gibco, ThermoFisher Scientific, sterile filtered using a 0.1 µm filter and stored at 4 °C) at 37 °C and 5% CO₂. Cell feeding is done as follows: 3–4 hours after the cells are plated, they are fed with 1000 µl of NGM. The cells are fed with 500 µl of NGM three days later. Four days later, 300 µl of the media is removed from the dish and replenished with fresh 500 µl of NGM. Then, every 3–4 days, 300 µl is removed, and 500 µl is added until they are transfected. Transfections were performed 12–15 days after plating by magnetofection using Combimag (OZ biosciences) and Lipofectamine 2000 (Life Technologies) according to the manufacturer's instructions.

For electroporation, the dissociated hippocampal cell suspension was obtained, as explained above, and neurons were electroporated using the Amaxa P3 primary cell 4D Nucleofector kit (Lonza) according to the manufacturer's instructions. Each electroporation reaction (per electroporation cuvette) constituted 750,000 cells and 2 µg of APEX-OMM DNA. 250,000 non-electroporated cells were later added to the electroporated cells to give 1,000,000 cells and were plated on poly-D-lysine coated 6-cm plastic dishes (Greiner). Cultures were maintained, as explained above.

### APEX biotin labeling
Neurons co-transfected with either APEX-Matrix or APEX-OMM, along with EGFP-Mito, as a mitochondrial marker, were used for these experiments between 13 and 15 DIV. For biotin labeling, neuronal media was replaced with either 500 µM biotin phenol (Berry and Associates, 500 mM stock made in DMSO) or 500 µM biotin (500 mM stock made in DMSO stored at 4 °C) in NGM for 30 min at 37 °C. Neurons were treated with 1 mM H₂O₂ (Alfa Aesar, 100 mM stock made in water and stored at 4 °C) for 1 min. The reaction was quenched by quencher solution consisting of (in mM) 10 Trolox (Merck), 20 sodium ascorbate, and 20 sodium azide (Roth) in DPBS, pH 7.4 adjusted with NaOH (filtered and stored at 4 °C), Neurons were subsequently washed thrice in either quencher wash solution consisting of (in mM) 5 Trolox, 10 sodium ascorbate, 10 sodium azide in DPBS for immunocytochemistry analyses or quencher solution for isolation of the biotinylated proteome.

### Immunostaining
Neurons were fixed in PFA (4% PFA in PBS) for 10–15 min, permeabilized (0.5% TritonX-100 in PBS for 30 min), and blocked in blocking buffer (4% goat serum in PBS for 1 h). For VAP immunostaining alone, neurons were fixed in PFA (4% PFA in Cytoskeleton Buffer (CB) – in mM, 10 MES, pH 6.1, 138 KCl, 3 MgCl₂, 2 EGTA, 0.3 M sucrose[63]) for 15 min, permeabilized (0.25% TritonX-100 in PBS for 15 min), and blocked in blocking buffer (2% BSA in PBS for 1 h). Fixed and permeabilized neurons were incubated with anti-biotin rb (1:5000, Bethyl, A150109A) for biotin labeling detection, anti-GFP chk (1:2000[6], AVES 1020) for EGFP-Mito immunoamplification, anti-V5 ms (1:500[32], Invitrogen R960-25, previously 460705, SV5-Pk1) for APEX-Matrix immunoamplification, anti-FLAG ms (1:500[32], Sigma F3165, M2) for APEX-OMM immunoamplification, anti-Map2 rb (1:100[6], Abcam, ab32454) as a dendritic marker, anti-VAP A/B ms (1:100, NeuroMab 75-496, N479/107) or anti-Vapa rb (1:100, Proteintech 152751AP), anti-Vapb rb (1:100, Proteintech 144771AP) for VAP immunoamplification, and anti-Snca ms (1:1000, BD biosciences 610787, 42/alpha-Synuclein) for Snca immunoamplification for 90 min. Neurons were washed in PBS, and fluorophore-coupled secondary antibodies anti-rb Alexa 405 (1:1000, Invitrogen A31556), anti chk Alexa 488 (1:1000, Invitrogen A11039), anti-ms Alexa 546 (1:1000, Invitrogen A11030), anti-ms Alexa 647 (1:1000, Invitrogen A21236), and anti-rb Alexa 488 (1:1000, Invitrogen A11008) were incubated for 45–60 min to detect biotin labeling, EGFP, APEX, VAP/Snca, and Vapa/b respectively. Immunostained neurons were either stored at 4 °C or imaged directly.

### Biotinylated proteome isolation
Neurons were electroporated with APEX-OMM to achieve higher transfection efficiency. Following biotin labeling on 14 DIV neurons, as explained above, cells were scraped. For each experimental batch of neuronal culture preparation expressing APEX-OMM, cells from up to

three 6 cm plastic dishes (adding up to 3,000,000 cells) were pooled together (to achieve ~70–100 µg of total protein). Cells were centrifuged at 70 × g for 10 min at 4 °C, and the pellet was either stored at −80 °C or continued with cell lysis. Cell pellets were suspended in RIPA lysis buffer (RLB) consisting of (in mM) 50 Tris, 150 NaCl, 1 PMSF, 10 sodium azide, 10 sodium ascorbate, 5 Trolox, 0.01% SDS, 0.5% sodium deoxycholate, 1% Triton X-100 and 1X protease inhibitor cocktail in water, pH 7.4 (filtered and stored in 4 °C) and incubated in ice for 20 min. Lysed cells were centrifuged at 15800 g for 10 min at 4 °C, and the protein supernatant was collected. Protein concentration was determined using Pierce 660 nm protein assay (Thermofisher) per the manufacturer's instructions. Protein samples were stored at −80 °C.

### Streptavidin enrichment of biotinylated proteome

Biotinylated proteins were affinity-purified using streptavidin-coated magnetic beads (Pierce) as follows: Beads were washed twice in RLB, and protein samples were incubated with the beads (at 18:1 bead-to-sample ratio) for 1 h at RT with gentle rotation (the flow through obtained here was used as FT). Beads were washed twice in RLB (the first wash was used as Wash), once in 2 M urea in 10 mM Tris-HCl, pH 8, and twice again in RLB. Beads were then washed thrice in RIPA wash buffer (RWB) containing (in mM) 50 Tris, 150 NaCl, 10 sodium azide, 10 sodium ascorbate, 5 Trolox, 0.1% SDS, 0.2% sodium deoxycholate and once with RWB2 containing (in mM) 50 Tris, 150 NaCl, 10 sodium azide, 10 sodium ascorbate, 5 Trolox and further processed for mass spectrometric analyses.

For western blot analyses, biotinylated proteins were eluted from the streptavidin beads using 3X NuPAGE LDS sample buffer supplemented with 20 mM DTT and 2 mM biotin and heating the mixture at 95 °C for 5 min. The elution step was performed twice to obtain Elu1 and Elu2 samples. Protein samples from input, FT, Wash, Elu1, and Elu2 were separated by electrophoresis (NuPAGE 4–12% Bis-Tris gel (Invitrogen)) and immunoblotted with anti-biotin ms (Sigma B7653, 1:2000) for detection of biotin labeling or stained with Coomassie for detection of total protein. Fluorophore-coupled secondary antibody anti-ms IR 800 (Invitrogen 92632210 LICOR, 1:10,000) was used for immunodetection.

### Mass spectrometric analysis

Streptavidin-bound biotinylated proteins were digested on-bead using an optimized protocol[64,65]. In brief, beads were washed 4 times in 1× PBS and twice in 50 mM ammonium bicarbonate, resuspended in 50 mM ammonium bicarbonate, and heated at 70 °C for 2 min with gentle agitation. 3 M urea was added, and the mixture was cooled down to RT. Bead suspension was reduced with 3.125 mM TCEP (0.5 M stock prepared in LC-MS-grade water) for 30 min at RT with constant agitation and incubated with 11.2 mM iodoacetamide (0.5 M stock prepared in LC-MS-grade water) for 30 min at RT in the dark with continuous agitation. Protein samples were digested sequentially with Lys-C and trypsin on Microcon-10 filters (Merck Millipore, #MRCPRT010 Ultracel YM-10). The mixture was centrifuged at 1000 × g for 5 min, and the supernatant containing peptides was collected. Digested peptides were desalted using ZipTips (Merck) according to the manufacturer's instructions, dried in a Speed-Vac, and stored at −20 °C. For mass spectrometry analyses, the dried peptides were reconstituted in 5% acetonitrile with 0.1% formic acid and loaded using a nano-HPLC (Dionex U3000 RSLCnano or Bruker NanoElute) on reversed-phase columns (trapping column: particle size = 3 µm, C18, L = 2 cm; analytical column: PepMap (Dionex, Thermofisher: particle size = 2 µm, C18, L = 50 cm) or PepSep (ReproSil-Pur C18-AQ, Marslev: particle size: 1.8 µm, C18, L = 50 cm)). Peptides were eluted using 90-minute (Dionex) or 120-minute (Bruker) gradients (buffer A: water with 0.1% formic acid) and acetonitrile (buffer B: acetonitrile with 0.1%

formic acid; all solvents purchased from FluKa; gradients were ramped from 2% to 35% B at flowrates of 300 nL/min) and analyzed using an Orbitrap Elite, Q Exactive Plus (Thermofisher) or Bruker Impact-II mass spectrometer.

On Orbitrap instruments, a full MS scan from m/z 350 to 1600 was acquired at a resolution of 120,000, and sequence information was acquired by computer-controlled, data-dependent automated switching to MS/MS mode using collision-induced dissociation (CID) fragmentation in the linear Ion Trap (IT-MS, Orbitrap Elite) or the Orbitrap (FTFT, Q Exactive Plus). In the Bruker Impact-II, full scans from m/z 150 to 2200 were recorded, and sequence information was acquired by a computer-controlled, data-dependent method with a fixed cycle and an intensity-dependent acquisition speed for MS/MS spectra between 8 and 20 Hz. Next, MSMS-spectra were searched against the protein database for Rattus norvegicus (TaxID 10116, Uniprot) and additionally for a database containing common MS-contaminations with trypsin as a digestion enzyme using MaxQuant (version 1.6.0.1) and Proteome Discoverer (PD) (Thermo Fisher, version 2.1). The set of stringent search constraints allowed tryptic peptides with up to two missed cleavages, a minimum of two valid peptides, a precursor mass tolerance of 10 ppm, and a fragment mass tolerance of 0.5 Da (Orbitrap Elite), 20 ppm (Q Exactive Plus) or 0.05 Da (Impact-II). In addition, carbamidomethylation of cysteine was set as a fixed modification, acetylation of peptide N-termini, oxidation of methionine, and biotin-labeling of tyrosine, tryptophan, and histidine were set as variable modifications. Percolator node, a machine-learning tool in PD 2.1, was used to estimate the number of false positive identifications. A high confidence q-value of less than 0.01 (False discovery rate (FDR) < 0.01) was assigned to filter both the peptide spectrum match (PSM) results and peptide results.

The procedure was repeated thrice, and three technical replicates were combined. The technical overlap for high confident proteins was determined, where at least two peptides from Rattus norvegicus were needed for identification. Proteins found in the Control sample were excluded from the experimental samples. Proteins found in at least two of the three replicates were considered for further analysis. The Gene/product IDs for these proteins were searched through Gene Ontology Resource (July 2, 2018 (2022-11-03)) for 'mitochondrion' and (June 11, 2023 (2023-06-11)) for 'endoplasmic reticulum' as GO cellular components with Rattus norvegicus as a filter. Overlap between the two searches was performed to find Gene/product IDs with 'mitochondrion' and 'endoplasmic reticulum' as GO annotations.

The Gene/product IDs GO annotated as 'mitochondrion' were further filtered for actin and tubulin interactors using a database for protein interactions, BioGRID, using the search term 'Actb' for actin interactors and 'Tuba1a' for tubulin interactors with Rattus norvegicus and Mus musculus as organism filters. Since the BioGRID 'Actb' search also resulted in actin-mRNA interacting proteins (Ddx3y, Matr3, Pcbp2), they were excluded from our list as we were only interested in actin-protein interacting proteins.

To identify the most abundant soluble proteins in our OMM proteome, first, the published neuropil proteome with a list of Gene/product IDs and their mean protein levels (log2 iBAQ values), obtained by the same methodological approach as our proteome dataset, was used[44]. While the digests were different (the samples in Biever et al.[44] were generated from lysed tissue, and our samples were produced via affinity purifications), the LC-MS was very similar regarding chromatography columns, setups, and acquisition strategies. Then, from the neuropil proteome list, the Gene/product IDs with 'membrane' as GO cellular component and Rattus norvegicus as a filter were eliminated to get a list of soluble proteins and their corresponding log2 iBAQ values. Overlaps between the first 200 soluble proteins with the highest iBAQ values (or all soluble proteins) and 129 proteins in the OMM proteome were then performed.

## Imaging

Live cell imaging was conducted 15-19 days after neuronal cell culture plating. All experiments, unless specified, were performed at 32 °C in E4 imaging buffer containing (in mM) 120 NaCl, 3 KCl, 10 HEPES (buffered to pH 7.4), 3 CaCl$_2$, 1 MgCl$_2$, and 10 Glucose. For spine plasticity induction by glutamate uncaging experiments, a modified E4 buffer lacking MgCl$_2$ with 4 mM CaCl$_2$ was used –the rest of the constituents remained unchanged. Airyscan confocal experiments were done at room temperature.

## Mitochondria-actin interaction measurements using Airyscan

Neurons were transfected with Fis1-Lifeact-GFP and Fis1-mCherry plasmid constructs, along with either Control shRNA or shRNA targeting a gene of interest (*CAP1*, *IMMT*, *SNCA*, *CYFIP1*, *NCKAP1*, *PFN2*, *SRGAP2*, *VAP*, or *MCU*). Transfected neurons were identified by Fis1-mCherry signal targeting mitochondria to differentiate between dendritic (long mitochondria) and axonal (short mitochondria) regions. For negative control experiments, neurons transfected with Fis1-Lifeact-GFP and Fis1-mCherry plasmid constructs were imaged either in the absence or presence of 25 µg/ml cytochalasin-D for 15 min (12.5 mg/ml stock made in DMSO), 1 µg/ml Latrunculin-B for 10 min (1 mg/ml stock made in DMSO), or 10 µg/ml nocodazole for 10 min (5 mg/ml stock made in DMSO). Neurons were transfected with Lifeact-mCherry and Mito-BFP in the absence or presence of Fis1-Lifeact-GFP to visualize any effect of Lifeact-GFP expression on actin morphology in dendrites. Neurons were also transfected with Lifeact-mCherry and Mito-BFP in the presence of Control shRNA or VAP shRNA, or the presence of Control Cas9 plasmid lacking sgRNAs or with the VAP CRISPR-Cas9 KO plasmid with two sgRNAs targeting *VAPA* and *VAPB*, respectively, to visualize any effect of VAP KD or KO on actin and mitochondria in dendrites and axons. The sequence of the *VAPA* sgRNA is 5' AGACAGTCACGATTGACCCT 3' and *VAPB* sgRNA is 5' CTGCGTGCGGCCCAACAGTG 3'[66].

Imaging was performed using an LSM 880 AxioObserver confocal microscope (Zeiss) with a Plan-Apochromat 63x/1.4 DIC oil immersion objective (Zeiss) and the Airyscan (GaAsP-PMT) detector. Images were acquired in 16-bit mode using unidirectional scanning, with a field of view of 66.45 µm × 66.45 µm (zoom 2.0), voxel sizes of (x,y,z) 40 nm × 40 nm × 800 nm, and pixel dwell times of 2.65 µs. The height (along the optical axis) of the volumetric z-stack was chosen to encompass the entire thickness of the neuron. Dual-color Airyscan imaging was performed sequentially using frame steps in Airyscan-SR mode with a default pinhole size set to 2.77 AU (for 488 nm excitation) and 2.5 AU (for 561 nm excitation). Laser power and detector gain were adjusted for each channel to utilize between 35% and 50% of the full dynamic range of the detector, carefully avoiding detector saturation. Typical laser powers were around 5.0% (488 nm) and 4.5% (561 nm) with a master gain set to 750 V. Raw Airyscan images were subsequently processed using Zeiss Airyscan processing set to default deconvolution strength settings. Imaging conditions were kept constant within a batch of experiments.

The Imaris software package (version 9.5.1; Oxford Instruments/Andor) was used to analyze the processed Airyscan images of the mitochondria-actin interaction experiments. First, two different surfaces were created, one for the mitochondria, red color channel (Fis1-mCherry), and one for the actin, green color channel (Fis1-Lifeact-GFP). Then, a colocalized surface was made in yellow using the 'colocalization' feature. Next, total volume was obtained from the red channel surface (total mitochondria) and the yellow colocalized surface (mitochondrial region interacting with actin) using the 'detailed statistics' feature. Finally, the percentage interaction between mitochondria and actin was calculated by dividing the volume of the yellow colocalized surface (mitochondrial region interacting with actin) by the volume of the red channel surface (total mitochondria).

For mitochondria-actin correlation coefficient analysis in cell bodies, three ROIs were manually selected per cell body in the mitochondria, red color channel (Fis1-mCherry). The same ROIs were used for the corresponding actin, green color channel (Fis1-Lifeact-GFP), for the same field of view. A mitochondria mask was generated using the selected ROIs and default auto thresholding. The Image J plugin Coloc 2 with a mitochondria mask and 4.27 PSF was used to calculate the Pearson correlation coefficient between mitochondria and actin.

## Optical measurements

Live imaging was performed using a custom-built inverted spinning disk confocal microscope (3i imaging systems; model CSU-W1) attached to an Andor iXon Life 888. Image acquisition was controlled by SlideBook 6 software. Images were acquired with a Plan-Apochromat 63x/1.4 NA. Oil objective, M27 with DIC III prism, using a CSU-W1 Dichroic for 488/561 nm excitation with Quad emitter and individual emitters, at laser powers 1.1 mW (488 nm) and 0.8 mW (561 nm) for photoactivation and Vapb OE (Vapb-emerald) experiments; and laser powers 2 mW (488 nm) and 2.7 mW (561 nm) for spine structural plasticity measurements. During imaging, the temperature was maintained at 32 °C using an Okolab stage top incubator with temperature control.

## Spine actin dynamics measurement

Neurons were transfected with LifeAct-mCherry and Mito-BFP in the absence or presence of Fis1-Lifeact-GFP to visualize any effect of Lifeact-GFP expression on spine actin dynamics, used as a proxy measurement for general actin dynamics in neurons. Neurons were also transfected with Lifeact-mCherry and Mito-BFP in the presence of Control Cas9 plasmid lacking sgRNAs or with the VAP CRISPR-Cas9 KO plasmid with two sgRNAs targeting *VAPA* and *VAPB*, respectively to visualize any effect of VAP KO on spine actin dynamics. Neurons were imaged in a modified E4 buffer lacking MgCl$_2$, the same as in spine structural plasticity measurements (see above). Images were acquired every ~15 s for up to 2 min, the same duration as the spine plasticity induction protocol (see below).

For analysis, the Weighted Center of Mass (CoM) for each spine over time was calculated using the Dendritic Filopodia Motility Analyser ImageJ plugin[67]. Spine actin displacement was calculated as the Euclidean distance between the CoM at the 2-min time frame with respect to the first-time frame.

## Photoactivation experiments and analysis

A custom-written automated program was used for simultaneous imaging of up to 10 neurons per dish, with a z-stack interval of 1 µm spanning a total of 5 µm, with both 488 nm and 561 nm lasers at exposure times of 15 ms. Neurons were transfected with Mito-PAGFP and Mito-DsRed plasmid constructs and Control shRNA or shRNA plasmids targeting the desired gene (*VAPA/B*, *SNCA*, or *SRGAP2*). Transfected neurons were identified using Mito-DsRed fluorescence. Photoactivation was performed using a multiphoton laser set-up at 805 nm (Chameleon, Coherent) and a Pockels cell (Conoptics) to control the photoactivation pulse. A 5×2–10×2 µm$^2$ photoactivation spot was chosen at a dendrite. Images were acquired every 30 s for 70 min. Following acquiring prephotoactivation fluorescence images for the first 6 time frames (3 min), a photoactivation pulse was given at the 7th time frame at 1 ms duration per pixel and 3.5–5.8 mW power and the imaging continued.

To measure the photoactivated mitochondrial compartment length from these experiments, images were converted to maximum intensity projections and analyzed by Imaris 8.4.1. Using the 'surface segmentation wizard', the photoactivated mitochondrial region was selected by setting a manual threshold, excluding the non-photoactivated background. The 'compartment length' was defined as the photoactivated mitochondria extracted across all the time

points using the 'filament wizard' and the subsequent addition of the photoactivated mitochondrial filaments detected above the set threshold during the imaging period.

For measuring the photoactivated mitochondrial compartment fluorescence from these experiments, an ROI of the size of the photoactivated mitochondrial compartment at the 5-min timepoint was drawn, and the fluorescence intensity was measured over the 60-min time course of the experiment. The photoactivated compartment stability index was calculated by dividing the photoactivated compartment fluorescence intensity at the end of 60 min post-photoactivation ($F_{60min}$) by the photoactivated compartment fluorescence intensity 5 min postphotoactivation ($F_{5min}$), resulting in $F_{60min}/F_{5min}$. A reduced compartment stability index statistically significant from the Control was considered an unstable mitochondrial compartment.

## Mitochondrial length and density analysis

Neurons were transfected with Mito-EGFP and Homer2-mOrange constructs along with Control Cas9 plasmid lacking sgRNAs or with the VAP CRISPR-Cas9 KO plasmid with two sgRNAs targeting *VAPA* and *VAPB*, respectively. Neurons were imaged in a modified E4 buffer lacking MgCl$_2$, the same as in spine structural plasticity measurements (see above). Homer2-mOrange fluorescence was used to identify spines. Dendritic segments with spines that were, on average, 110 μm away from the cell body were chosen for imaging, same as in spine structural plasticity measurements (see below). A z-stack of 11 planes with a step size of 0.5 μm was acquired.

For analysis, maximum intensity projection of the z-stacks was done. Dendritic segments up to 45 μm length on either side of the spine were chosen for further analysis. Mitochondrial length and density were measured using a custom ImageJ/Fiji macro script. Mitochondria were manually traced using the 'Segmented Line' tool for measuring mitochondrial length. In cases where a mitochondrion extended out of the field of view, and only a partial mitochondrion could be imaged, its length was still measured as the minimal measurable length and combined with the rest of the mitochondrial lengths in the summarized data. So, the average mitochondrial length in the summarized data might be an underestimation, particularly in Control, where mitochondria can be as long as ~30 μm[17]. For mitochondrial density analysis, the same mitochondrial traces were used, and the dendrite was partitioned into 15 μm segments at distances 0–15, 15–30, and 30–45 μm on either side of the spine of interest. The length of each mitochondrial trace within a dendritic segment was added and normalized to the 15 μm length to get mitochondrial density.

## shRNA knockdown efficiency determination

Following photoactivation or spine structural plasticity measurements, the neurons were fixed and immunostained, as explained above. Expression levels were quantified by selecting soma regions of interest (ROIs) located by either GCaMP signal (from structural plasticity measurements) or Mito-PAGFP signal (from photoactivation experiments) followed by measuring VAP, Snca, or any other protein's immunofluorescence intensity at the same ROIs after background subtraction. The average VAP or Snca fluorescence intensities measured in GCaMP-positive or Mito-PAGFP-positive, transfected somas ($F_{trans}$) and the average VAP or Snca fluorescence intensities measured in GCaMP-negative or Mito-PAGFP-negative somas from adjacent untransfected neurons ($F_{untrans}$) were obtained from the same illumination field. The knockdown percentage was calculated as $F_{trans}/F_{untrans}$.

## Correlation coefficient measurements

Neurons transfected with Mito-DsRed were fixed and immunostained for Vapa or Vapb, as explained above. Images were acquired with an LSM 880 confocal microscope (Zeiss) using a C-Apochromat 40x/1.2 W objective and a pinhole size of 1 airy unit. Images were acquired in 16-bit mode as Z stacks, with a pixel pitch of 208 nm and field of view of 212.55 × 212.55 μm$^2$, throughout the entire thickness of the neuron with an optical slice thickness of 0.5 μm and pixel dwell times of 0.5 μs. The detector gain in each channel was adjusted to cover the full dynamic range, avoiding saturated pixels. Imaging conditions were kept constant within a batch of experiments. Maximum intensity projections were used for image analyses.

For live imaging, neurons were transfected with Mito-DsRed and Vapb-emerald or Vapb-HaloTag. For Vapb-HaloTag experiments, neurons were incubated with 200 nM JF635 HaloTag dye (Janelia, 200 μM stock made in DMSO) for 15 min and washed three times in E4 buffer. Images were acquired from the dendrites and axons using the spinning disc confocal microscope, as explained above.

For correlation coefficient analysis, dendritic and axonal regions of interest from both immunostained and live neuron images were straightened from the respective Vapa/b (green) and mitochondria (red) channels using ImageJ. A line of ~3 μm thickness was drawn along the straightened dendrite or axon from the respective channels, and a line profile was obtained. Each dendrite or axon's Vapa/b line profile was plotted against its mitochondria line profile, and the corresponding Pearson correlation coefficients were determined using OriginPro 2022 (64-bit) SR1. For shuffled control (Shf), one of the line profile measurements (mitochondria channel) was shuffled, while the corresponding line profile pair (VAP channel) remained undisturbed, and the corresponding Pearson correlation coefficient was determined.

## Spine structural plasticity measurements

Neurons were transfected with GCaMP6s, Mito-DsRed, and PSD95-mCherry or Homer2-mOrange plasmid constructs, along with Control Cas9 plasmid lacking sgRNAs or with the VAP CRISPR-Cas9 KO plasmid with two sgRNAs targeting *VAPA* and *VAPB*, respectively. Transfected neurons were identified by a change in GCaMP6s fluorescence corresponding to calcium transients in dendrites and spines. In addition, PSD95-mCherry or Homer2-mOrange fluorescence was used to identify spines for stimulation by two-photon glutamate uncaging or to identify adjacent spines in the same dendritic segment or sister dendrites. We did not differentiate between mushroom, stubby, or other spine shapes. Before glutamate uncaging, neuronal imaging buffer was replaced with 1 μM TTX (Citrate salt, 2 mM stock made in water), 50 μM Forskolin (Tocris Bioscience, 25 mM stock made in DMSO), and 2 mM 4-Methoxy-7-nitroindolinyl-caged-L-glutamate (MNI caged glutamate) (Tocris Bioscience, 100 mM stock made in E4 buffer) in modified E4 buffer lacking MgCl$_2$ (see above). Glutamate uncaging was performed using a multiphoton laser 720 nm (Mai TAI HP) and a Pockels cell (ConOptics) to control the uncaging pulses. Distal spines or distal dendritic segments that were, on average, 110 μm away from the cell body were chosen for uncaging experiments (as we use dissociated cultures, we could not differentiate between apical and basal dendrites). Spines with at least one mitochondrion within 3 μm from the spine's base were chosen for plasticity experiments. However, the length of the mitochondrial compartment was not predefined. To test a spine's response to an uncaging pulse, an uncaging spot (2 μm$^2$) close to a spine-head was selected, and two to three uncaging pulses at 10 ms pulse duration per pixel and 7.0–9.4 mW power were given and checked for spine-specific calcium transients. An uncaging protocol of 60 uncaging pulses at 0.5 Hz with 10 ms pulse duration per pixel at 7.0–9.4 mW power was used. Images were acquired before plasticity induction at $t = 0$ min, at the end of the plasticity induction at $t = 2$ min, and then every 10 min for up to 62 min at $t = 12, 22, 32, 42, 52, 62$ min. Up to 3 spines and their corresponding dendrites were studied per neuron. Each spine was counted as a single experiment, and the spines were averaged to get the final data summary.

Following structural plasticity measurements, neurons were fixed in CB buffer as explained above, stored either for 3–4 h at room temperature (or at 4 °C overnight), and continued with the VAP immunostaining protocol (see above).

For analysis, ten images from each timepoint were averaged, and the background was subtracted using a rolling ball filter of a radius 50 pixels using FIJI. Next, a line crossing the center of the spine-head was drawn using a custom-written MATLAB script[6]. Then, the fluorescence intensity measured along the line was fit to a Gaussian to obtain the full width at half maxima (FWHM) –defined as the spine-head width[68]. As the FWHM is independent of fluorescence intensity, it measures spine-head width even from fluctuating GCaMP fluorescence in spines. Once the plasticity-induced spine was analyzed for its spine-head width, the adjacent spines on either side of the plasticity-induced spine, within the 45 μm distance, were used for analysis. A few spines (16 out of 59 spines, color-coded in black in Fig. 6f, g, Supplementary Fig. 6e) did not have any adjacent spines either because the adjacent spines were not in the imaging focal plane or because of the low spine density of the neuron, or because the plasticity-induced spine was positioned close to the edge of the imaging field of view.

### ER calcium measurements and analysis

Neurons were transfected with ERGCaMP210 and Homer2-mOrange plasmids, along with Control Cas9 plasmid lacking sgRNAs or with the VAP CRISPR-Cas9 KO plasmid with two sgRNAs targeting *VAPA* and *VAPB*, respectively. Transfected neurons were identified by ERGCaMP210 fluorescence, and Homer2-mOrange fluorescence was used to identify spines. Two-photon glutamate uncaging was conducted as described above. To test ER calcium release on spine stimulation, an uncaging protocol of 6 uncaging pulses at 0.2 Hz with 100 ms pulse duration per pixel at 7–9.3 mW power was used. To induce synaptic plasticity, an uncaging protocol of 59 uncaging pulses at 0.5 Hz with 10 ms pulse duration per pixel at 7–9.3 mW power was used.

For image analysis, ERGCaMP210 fluorescence was used to draw a 1–2 μm long ROI on ER at the base of the plasticity-induced spine using the "Wand tool" in ImageJ. The selected ROI was used to measure ER calcium from the ERGCaMP210 channel. All measured fluorescence intensities were background subtracted and converted to $\Delta F/F_0$ calculated as $(F-F_0)/F_0$, where $F_0$ is the average of baseline fluorescence before plasticity-induction, and F is the fluorescence intensity at a particular time point post-plasticity induction. For statistics, peaks of the $\Delta F/F_0$ time trace were detected using the "find peaks" function under the "peak analyzer" tool in Origin with the "Local Maximum" method. The baseline mode was set to "None (Y = 0)". The direction of the peaks was specified as "negative" and the number of local points was set to 1. The average peak values from Control and VAP KO dendrites were compared.

### Mitochondrial calcium measurements and analysis

Neurons were transfected with mitoGCaMP6f and RCaMP1.07 plasmids, along with Control Cas9 plasmid lacking sgRNAs or with the VAP CRISPR-Cas9 KO plasmid with two sgRNAs targeting *VAPA* and *VAPB*, respectively. Transfected neurons were identified by a change in RCaMP1.07 fluorescence corresponding to calcium transients in dendrites and spines. Two-photon glutamate uncaging was conducted as described above. To induce synaptic plasticity, an uncaging protocol of 59 uncaging pulses at 0.5 Hz with 10 ms pulse duration per pixel at 5.6–8.2 mW power was used.

For image analysis, mitoGCaMP6f fluorescence was used to draw a 2 μm long ROI on mitochondria at the base of the plasticity-induced spine using the "Wand tool" in ImageJ. The selected ROI was used to measure mitochondrial calcium from the mitoGCaMP6f channel and dendritic calcium from the RCaMP1.07 channel. For spine calcium measurement, a 0.8 μm² circular ROI was drawn around the plasticity-induced spine and measured in the RCaMP1.07 channel. All measured fluorescence intensities were background subtracted and converted to $\Delta F/F_0$ calculated as $(F-F_0)/F_0$, where $F_0$ is the average of baseline fluorescence before plasticity-induction, and F is the fluorescence intensity at a particular time point post-plasticity induction. For statistics, peaks of the $\Delta F/F_0$ time trace were detected using the "find peaks" function under the "peak analyzer" tool in Origin with the "Local Maximum" method. The baseline mode was set to "None (Y = 0)". The direction of the peaks was specified as "positive" and the number of local points was set to 1. The average peak values from Control and VAP KO dendrites were compared.

### Quantification and statistical analysis

Statistical details of experiments, including statistical tests and n and *p* values, can be found in the figure legends. Each animal corresponds to one weekly batch of neuronal culture preparation. In all the figures, the box whisker plots represent the median (line), mean (point), 25th-75th percentile (box), 10th-90th percentile (whisker), 1st–99th percentile (X), and min-max () ranges. Error bars in bar graphs and line traces are SEM. A *p*-value of less than 0.05 was considered significant for all statistical tests.

No statistical method was used to predetermine the sample size. Sample sizes were similar to or larger than those reported in the previous publications in the field and sufficient for our claims based on statistical significance.

### Software and code

The image and data analysis software used in this study are MATLAB (R2017b and R2019b), ImageJ 1.52 S, IMARIS 8.4.1 and 9.5.1, Microsoft Excel 16.69.1, OriginPro 2022 and R 4.3.1.

### Reporting summary

Further information on research design is available in the Nature Portfolio Reporting Summary linked to this article.

## Data availability

The mass spectrometry proteomics data generated in this study have been deposited in the ProteomeXchange Consortium via the PRIDE partner repository[69] under accession code PXD047226 (https://www.ebi.ac.uk/pride/archive/projects/PXD047226). The raw imaging data will be provided by the corresponding author upon request. The source data generated in this study are provided in the Source Data file. Source data are provided with this paper.

## Code availability

Custom-written scripts used for data analysis are available at https://github.com/Rangaraju-Lab/Bapat-et-al-2023.git [70].

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

## Acknowledgements

We thank A.Y. Ting for the APEX plasmids, U. Manor for the Fis1-Lifeact-GFP, Fis1-mCherry, and Spire1 KD plasmids; M. Davidson for the Lifeact-mCherry plasmid; G. Voeltz for the Mito-BFP plasmid; D. Kashatus for the Drp1 KD plasmid; E. Burton for the Snca KD plasmid, F. Polleux for the Mff OE and Srgap2 KD plasmids; C. Hoogenraad and M. Hoppa for the VAP KD and KO plasmids respectively; H. Higgs for the Inf2 KD plasmid; R. Youle for the Mito-PAGFP plasmid; J. Lippincott-Schwartz for the Vapb-emerald and Vapb-HaloTag plasmids; C. Hanus for the ER-VSVG-GFP and PSD95-mCherry plasmids; and J. de Juan-Sanz for Homer2-mOrange plasmid. We are also thankful to E.M. Schuman for her support in the initial stages of this project; I. Wüllenweber for proteomics and mass spectrometry support; S. Perez, I. Bartnik, N. Fuerst, and A. Staab for technical assistance; L. Yan for assistance with the Optical Workshop Core; N. Urban for assistance with the MPFI Light Microscopy Core; R. Yasuda, H. Inagaki, and all members of the Rangaraju Lab for critical input on the manuscript. V.R. is funded by the Max Planck Society, and T.P. was funded by the Nambu Summer Research Undergraduate Scholarship. This project has also received funding from the European Union's Horizon 2020 research and innovation program under the Marie Sklodowska-Curie grant agreement 657702 and an EMBO Long-Term Fellowship 898-2014 for V.R. and the European Research Council grant agreement 743216 for E.M. Schuman.

## Author contributions

V.R. designed experiments. Experiments were carried out by O.B., T.P., S.K., R.F., C.T., F.R., and V.R. Data were analyzed by O.B., T.P., S.K., M.S., R.F., F.R., and V.R. J.D.L. supervised mass spectrometry data acquisition and analysis, and V.R. supervised the rest of the experiments and analysis. V.R. wrote the manuscript, and all authors edited it.

## Funding

## Competing interests

The authors declare no competing interests.
