## [Peer Review File · Nature Communications]

VAP spatially stabilizes dendritic mitochondria to locally support synaptic plasticityREVIEWER COMMENTS

Reviewer #1 (Remarks to the Author):

In this study, the authors utilized APEX2 proximity labeling to identify proteins interacting with the mitochondrial outer membrane, electing to study 8 proteins that are also known to interact with actin. They found that each of these reduced mitochondrial-actin contact sites in dendrites, but not in axons. Knocking down VAPA/VAPB, SNCA or SRGAP2 also reduced the length of stable mitochondrial compartments that the authors had described in a previous paper. Using glutamate uncaging, they state that depletion of VAPA/B did not prevent the short-term increase in spine size, but may prevent the field effect on adjacent spines. Moreover, the increased size of stimulated VAPA/B-deficient spines that was observed at 2 min was not sustained to 60 min as in control spines.

Identification of the ER protein VAPA/B as an important determinant of mitochondrial tethering to actin as well as regulating changes in spine size following glutamate uncaging is novel and interesting. Strengths include novel assays for actin-mitochondria association and use of some state-of-the-art technology. However, the data remain correlative and it is not clear that the changes in mitochondrial compartment size are causally linked to the spine size phenotypes, and a more careful, in-depth analysis is required to gain mechanistic insights.

1. Many of the main conclusions hinge on Fig. 6. However, there is a lack of detail concerning how the segments of dendrites were selected for study. Are these apical or basal dendrites? Proximal or distal? Was a single dendrite studied per neuron, or do the values reflect averages of multiple dendrites? Were defined mitochondrial compartments used to select the segment to analyze? Was the activated spine always proximal to the other spines studied as depicted in the schematic? Moving distally, there is often a decrease in mitochondrial density and/or membrane potential - how was this controlled for?
2. There is a disconnect between the length of the stable compartment defined morphologically, which drops to ~17 microns according to Fig. S6E, and the region of adjacent spines whose head size gains are proposed to be fueled by this compartment, as there are none in the 0-15 micron range. If compartment size is the determining factor in synchronizing spine head width responses in adjacent spines, one might expect at least a trend of spine head responses correlating with the now smaller compartment.
3. Also, Supplementary Figure S6A shows a statistically significant increase in spine size at baseline in the VAP depletion group. However, if VAP helps to stabilize the mitochondria near spines, it is unclear why VAP depletion would lead to this increase. Further investigation is needed to understand the relationship between VAP, mitochondrial stability, and spine size regulation.
4. Although the text states there are no basal differences in calcium influx between control and VAP KO, the graphs in Fig. 6B look strikingly different. More work is needed to understand the effects of VAP KO on spine head size and on calcium flux at baseline before considering activity-dependent changes. Is the assay already saturated at baseline for VAP KO?
5. What percentage of the spines at different distances from the stimulated spine are increasing in size vs. decreasing in size. Is there a greater percentage of spines getting smaller due the larger average size of VAPKO spines at baseline?
6. The potential roles of altered fission-fusion dynamics, mitochondrial turnover are not considered.
7. Is the inability to sustain increased spine size due to decreased mitochondrial content in general, and can this be replicated by other causes of energy depletion?
8. Does VAPA/B knockdown affect the percent of PSD95 or Homer positive spines? Could this account for the loss of co-stimulatory head changes in adjacent spines?
9. There is a lot of variability in the numbers of neurons and animals used, ranging from 30 neurons from 10 animals in the supplementary figure to only 4 neurons from 2 animals in several key figures in the main manuscript. A more detailed description of the number of neurons, number of animals for each group, and number of times the experiment was replicated is necessary to evaluate reproducibility. It would be helpful to color code the spines from different neurons or different animals in 6E, for example, to see if the variability in direction of spine size change from baseline segregates by animal or by neuron.
10. What is the p value for Control vs. MCU neg control in Fig. 3C

11. In Fig. 6C, is 60 min significantly different from 2 min in the VAPKO curve?
12. Further validation of the APEX2 strategy is needed. The system may be too promiscuous if the majority of APEX-OMM labeled proteins are soluble proteins that quickly diffuse away. In Fig. 1B, it would be helpful to see what happens after permeabilization of the plasma membrane to allow labeled cytosolic proteins to diffuse out. One would expect to see a pattern similar to 1C, verifying that some labeled proteins are stably associated with the OMM.
13. It is curious that the screen failed to detect any proteins involved in axonal mitochondrial interactions with actin; for example, FHL2 mentioned in the text was not in the supplementary table (and as pointed out VAPB itself was not identified). Can authors comment on what steps were taken to validate sensitivity for known OMM interacting proteins and specificity relating to proteins that may bump the OMM with no functional interaction?
14. In Fig. S4E – both XNCA and SRGAP2 show a greater effect on compartment length compared to VAP. What was rationale to focus on VAP? In Fig. 4, the SRGAP2 data may be underpowered to detect significant differences as this group has much fewer data points than the other groups.

Minor points:

15. Since all 8 proteins had effects on actin-mitochondria contacts, what was rationale to focus only on VAP, SNCA and SRGAP2?
16. Given differences in abundance that was raised as an explanation for why the APEX procedure did not identify VAPB, it would be interesting to see the relative abundance of VAPA and VAPB by western blot or RT-PCR in hippocampal cultures.
17. Authors should strive to utilize more consistent terminology that clearly indicates when both paralogs were knocked down or are being discussed together, such as VAPA/B. On line 143, it states that VAPA was knocked down, but line 190 it indicated that all experiments involved knockdown of both A and B, and in many places VAP is used.
18. In Fig. 5, are either of the VAP(s) associated with somatic mitochondria?
19. The main conclusion, which is repeated verbatim in several positions in the paper “VAP functions as a spatial stabilizer that temporally sustains synaptic plasticity for up to ~60 min and as a spatial ruler that determines the ~30 μ m spatial dendritic segment fueled by mitochondria during synaptic plasticity” is not actually investigated or supported by the data presented. As only the 60 min time point was examined, it is unclear whether an intermediate-sized ~17 micron compartment might sustain plasticity for a shorter period of time, or as mentioned in Point 2, shorten rather than abolish the region of responsive adjacent spines.
20. In the synaptic plasticity experiment, line 227 lists the early time point as 12 min. Is this a typo?

Reviewer #2 (Remarks to the Author):

This manuscript describes a role for VAP in tethering dendritic mitochondria to actin, and a potential effect on dendritic spines. They first conduct APEX-OMM to identify proteins near neuronal mitochondria, and then examine the ones which are implicated in the cytoskeleton. They then examine how 8 of these including VAP regulate actin around mito, and show that VAP KD then decreases actin around mitochondria, and also mitochondrial fluorescence loss over time. They further show that VAPB is enriched in dendrites, and finally show that VAP KO cells have decreased spine size after stimulation. Overall, the concept of identifying actin-related proteins near mitochondria which regulate dendritic mitochondria and spine size is very interesting. However, there are currently major gaps in the mechanisms and conclusions, which could be greatly improved by addressing the comments below.

Major comments:

1. Fig 2: The authors may want to check how proteins are classified (including their

roles/localization). For example, SNCA is not classically thought of as a cytoskeletal protein.

2. Fig 2: As VAPA is localized to ER, and a list of other ER-localized proteins found in their list would be helpful if the authors are interested in the role of the ER on mitochondria.
3. Fig 3: Authors may want to use a control of their Fis1 plasmid: ex. Does change in mito size (ex. Drp1 KD vs Mfn2 KD) alter the mito-actin interaction (%) measure?
4. Fig 3: Images of normal Lifact-mCherry with mitochondria, colocalized with their Fis1-Lifact-GFP (or in conditions without it) would be helpful, to confirm that their new plasmid is not abnormally disrupting actin dynamics or its enrichment near mitochondria.
5. Fig 3: It is also unclear if VAP KD disrupts all actin, or just actin around mitochondria, so images of normal Lifact throughout neurons upon either VAP KD or KO would be helpful.
6. Fig 3C: Does KD of candidate proteins such as VAP change mito-actin interaction (%) in cell bodies?
7. Fig S4: The authors should show KD efficiency by western blot (rather than by % decrease in immunofluorescence, which can be subject to changes in cell morphology, and secondary antibody fluorescent aggregates)
8. Fig 3: How does VAPA mechanistically tether actin to mitochondria? (Does it increase actin polymerization; or bind directly to actin; and /or mitochondria)?
9. Fig 4A: It is unclear if the decrease in mito fluorescence in all conditions is from bleaching, or from fission/fusion? Can the authors confirm this a different way?
10. Fig 5: These images are not very clear at this resolution. Showing perhaps a cytosolic and/or ER marker would be helpful to know the shape of the dendrites and axon (and to know whether VAPB is just filling the area or are just all over the ER, rather than specific to mito).
11. Fig S5: Can the authors show any live movies of VaPB-emerald with Mito-DsRed, to further show that the signals are moving together over time?
12. Fig 6: Do VAP KO cells also show similar effects on photoactivation loss (similar to Vap KD in Fig 3)? Are there changes in mitochondrial or actin dynamics (in general using normal Lifact) in these VAP KO cells?
13. Fig 6: What happens to mitochondrial dynamics/density/size/mitochondrial calcium and Fis1-lifact in VAP KO cells upon 2-photon uncaging? Do changes in mitochondria and actin correspond to changes in spine shape (in both activated and neighboring spines) in VAP KO cells? This would help to confirm that changes in spines is due to VAP's role in mitochondria/actin interactions, rather than through its function on the ER.
14. Fig 6: Does VAP localization alter upon 2 photon uncaging?
15. Can the authors also see if specific mutations in VAP (that might disrupt its function or ER localization) alter this ability to disrupt spines?
16. Multiple other proteins have previously implicated in the role of actin around mitochondria (ex. INF2: Korobova, Science 2013; Myosin II: Korobova, Curr Biol 2014; Spire1C: Manor, Elife 2015; Myo19: Coscia, J Cell Sci 2023). Do these proteins show up in the OMM-APEX? Can the authors see if VAP alters actin around mitochondria through changes in these proteins (such as altering their localization/dynamics/levels; or possibly, if KO of these proteins also changes their Fis1 actin probe, as a control)?

Minor comments:

1. Fig 1: It is unclear why the APEX-matrix is being shown. Was mass-spec done for hits on APEX-matrix as well, and if so, how are these different from APEX-OMM?
2. Fig 3A: List schematic as its own panel (not on the figure, which is a little hard to read).
3. Fig 3B: Zoom in examples would be extremely helpful.
4. Fig 4A: Kymographs need a scale bar for time (y axis); and also a marker indicating time of initial photoactivation (if shown on kymograph).
5. Fig 5A/B: Gray vs dotted gray arrows are a little hard to tell apart: perhaps use of different colors or types of arrowheads would be better.

Reviewer #3 (Remarks to the Author):

In this article, Bapat and colleagues investigate how mitochondria are tethered to the cytoskeleton and how this tethering in dendrites may be involved in synaptic plasticity. Using a range of molecular and imaging techniques they generate a biotinylation screening and identify several candidate proteins that tether dendritic mitochondria. Next, they investigate the role of vesicle-associated membrane associated protein (VAP) in more detail comparing axons and dendrites. They find that VAPB (but not VAPA) tethers mitochondria to the cytoskeleton in dendrites but not axons. Moreover, knocking out VAP impairs spine dynamics in protocols for inducing long-term synaptic plasticity. The molecular techniques are excellent and address an important research question, as it is becoming increasingly clear that mitochondria behave differently in various subcompartments of the cell. While most data are clearly described some of the conclusions lack supporting evidence.

Major points

1. The main conclusion the authors draw is that mitochondria 'fuel' synaptic plasticity, and that disrupting their tethering impairs their ability to provide sufficient ATP to support spine growth. This is alluded to in the introduction (line 65), in the paragraph title (line 205), the title of Figure 6 and in the manuscript title. However, there is no supporting data presented. Which fueling mechanisms are affected by the VAP KD/KO? Do mitochondria have a smaller 'range' (due to a smaller compartment, Fig. 6F), become more motile, is ATP production disrupted, or is there a role for mitochondrial calcium buffering? To examine fueling a critical piece of evidence is showing ATP production in control and VAP KO conditions, particularly as the authors point out in the discussion (line 326) that ER-MCSs are important for ATP production, and VAP is involved in ER-mitochondrial contact. This could be done by targeting an ATP sensor to mitochondria.

2. The authors imply that mitochondria are smaller upon VAP KO/KD (Figure 6F), but no evidence is presented. MitoTracker or mito-dsRed imaging is required to provide a more direct readout of mitochondria. These data are also essential to combine with the photoactivation and spine uncaging experiments, showing overlays of images, at high resolution. How do mitochondria position relative to the targeted spines? This data will be important to interpret the 30 μm range the authors refer to.

3. In Figure 6B the GCaMP fluorescence at baseline seems to be higher in the VAP KO. What's the explanation? Is the baseline spine morphology affected? The authors write that VAP KO resulted in a "modest and negligible" spine-head size increase (line 219). However, data in Figure S6A demonstrates a significant increase (two-sample t-test, $n = 27$ from 7 animals). Referring to statistically significant differences as "negligible" seems inappropriate. The VAP KO shows several large spines $>1.2 \mu\text{m}$ not observed in control raising the question whether this limits their capacity to grow during the plasticity paradigm, explaining the limited spatial gradient of plasticity. It would also be useful to see the absolute (non-normalized) data from the plasticity experiments in Figure 6.

4. Figure 5D is missing a critical comparison to support the author's conclusion that VAPB is enriched near dendritic but not axonal mitochondria. It compares VAPB against VAPA but we need statistical comparison of dendritic vs. axonal VAPB.

Minor points

- In general, the figures could be increased in clarity. Merged images are lacking, for example, Figure 1D/E, Figure 5A/B (bottom). Conversely, separate images for each channel should be added to Figure 3A/B. Also, even though it was published before, it would be useful to add a small cartoon of the method used in Figure 4 or at least explain it briefly in the main text. Figure 4A would also become clearer if the fields of view to which the kymographs correspond were added (like Figure 3A), ideally at $t = 0$ and $t = 60$ minutes. The line profiles in Figure 1D and E are missing a scale along the x axis. Figure 5A/B would benefit from line profiles like the ones used in Figure 1 and Figure S5.

- In Figure 4C SRGAP2 doesn't show a significant difference from control, but also has relatively

few datapoints. What is the power? If more data points were added this could show that SRGAP2 is also important (which the authors hint to in the paragraph title). Please add data or clarify.

- In Figure S2B the authors claim an equal amount of protein input was used, although the input for "APEX-OMM" is higher in comparison to "- APEX". Quantifications are needed. This part also needs to be included in the main figures, as it is critical evidence. Independent and better evidence for the localization of APEX to the IMM and OMM could be provided, for example using ultrastructure electron microscopy.

- It is unclear if ER function is still normal after VAP disruption. ER also is involved in synaptic plasticity (see e.g. doi.org/10.1073/pnas.0905110106; doi: 10.1038/s41467-020-18889-5) and VAP is important for ER function as the authors point out. The authors should speculate on this or at least mention this caveat in the discussion.

- In Figure 6 supporting images at high resolution across a longer range of the dendrite is missing. These are shown in Fig. S6C but should be included in the main figure.

- The authors should better clarify the relationship of VAP with ALS or neurodegenerative diseases. They refer to ALS throughout the manuscript, even within the abstract (line 20) but don't make clear what the role of VAP and dendritic mitochondria tethering could have in these disorders.

- It is unclear which type of spines the authors included in their analysis (mushroom, stubby, ...). This should be clarified in the methods.

- In Fig. S5A the background of the dendrite channel seems very low (almost absent) compared to that of the axon in Fig. S5B. Was the same background subtraction used? Dendritic mitochondria are much longer than those in axons. Could this not explain why the authors find more colocalization with VAPB in dendrites compared to axons?

- Line 585, line 806 "neurons were replaced with ...", please rewrite.

We thank the reviewers for their thorough analysis of the manuscript and their many thoughtful suggestions. We have conducted many new experiments and analyses during the revision period, and we find the manuscript improved over the original submission. We briefly summarize the experiments conducted below, followed by a detailed point-to-point response to each reviewer.

Summary of new experiments conducted (in the order of appearance in the manuscript):

1. APEX OMM labeling after allowing for soluble proteins to diffuse out (attempted) (reviewer 1, question 12)
2. Confocal Airyscan static images of actin in the absence and presence of Fis1-Lifeact (**Supplementary Fig. 3a**) (reviewer 2, question 4)
3. Actin dynamics in the absence and presence of Fis1-Lifeact (**Supplementary Fig. 3b**) (reviewer 2, question 4)
4. Mitochondrial-actin interaction percentage on lengthening (using Drp1 KD) or shortening (using Mff OE) mitochondrial size compared to Control (**Supplementary Fig. 3c**) (reviewer 2, question 3)
5. Mitochondria-actin interaction percentage on knocking down mitochondrial fission proteins (Inf2 and Spire1C) compared to Control (**Supplementary Fig. 3c**) (reviewer 2, question 16)
6. Confocal Airyscan static images of actin in dendrites and axons of Control, VAP KD and VAP KO neurons (**Supplementary Fig. 3e, f**) (reviewer 2, questions 5, 12)
7. Actin dynamics in Control and VAP KO spines (**Supplementary Fig. 3g**) (reviewer 2, question 12)
8. Mito-actin interaction on 2p stimulation in Control and VAP KO neurons (attempted) (reviewer 2, question 13)
9. Mitochondria-actin interaction percentage in cell bodies in Control and VAP KD neurons (**Supplementary Fig. 3j**) (reviewer 1, question 18; reviewer 2, question 6)
10. Additional mitochondrial photoactivation experiments in Control and Srgap2 KD neurons (**Fig. 4c, d, Supplementary Fig. 4a, b**) (reviewer 1, question 14; reviewer 3, minor point 2)
11. Mitochondrial length measurements in Control and VAP KO neurons (**Supplementary Fig. 4c**) (reviewer 3, question 2)
12. Mitochondrial photoactivation experiments in Control and VAP KO neurons (**Supplementary Fig. 4d**) (reviewer 2, question 12)
13. VAPB localization with ER and mitochondria (**Supplementary Fig. 5f, g**) (reviewer 2, question 10)
14. VAPB-emerald imaging at the base of plasticity-induced spines (attempted) (reviewer 2, question 14)
15. Photoactivation and synaptic plasticity experiments with VAPB-P56S mutation (attempted) (reviewer 2, question 15)
16. Synaptic plasticity experiments in Control, Srgap2 KD, and Snca KD neurons (attempted) (reviewer 1, question 19)
17. ER calcium measurements in Control and VAP KO neurons (**Supplementary Fig. 6i**) (reviewer 3, minor point 4)
18. Mitochondrial calcium measurements in Control and VAP KO neurons (**Supplementary Fig. 6j**) (reviewer 1, question 7; reviewer 2, question 13; reviewer 3, question 1)
19. Mitochondrial content (density) measurements in distal dendrites in control and VAP KO neurons (**Supplementary Fig. 6k**) (reviewer 1, questions 1 and 7)
20. Mitochondrial and spine ATP measurements in Control and VAP KO neurons (data not included as ATP reporter is yet to be published) (reviewer 1, question 7; reviewer 2, question 13; reviewer 3, question 1)

Reviewer #1 (Remarks to the Author):

In this study, the authors utilized APEX2 proximity labeling to identify proteins interacting with the mitochondrial outer membrane, electing to study 8 proteins that are also known to interact with actin. They found that each of these reduced mitochondrial-actin contact sites in dendrites, but not in axons. Knocking down VAPA/VAPB, SNCA or SRGAP2 also reduced the length of stable mitochondrial compartments that the authors had described in a previous paper. Using glutamate uncaging, they state that depletion of VAPA/B did not prevent the short-term increase in spine size, but may prevent the field effect on adjacent spines. Moreover, the increased size of stimulated VAPA/B-deficient spines that was observed at 2 min was not sustained to 60 min as in control spines.

Identification of the ER protein VAPA/B as an important determinant of mitochondrial tethering to actin as well as regulating changes in spine size following glutamate uncaging is novel and interesting. Strengths include novel assays for actin-mitochondria association and use of some state-of-the-art technology. However, the data remain correlative and it is not clear that the changes in mitochondrial compartment size are causally linked to the spine size phenotypes, and a more careful, in-depth analysis is required to gain mechanistic insights.

1. Many of the main conclusions hinge on Fig. 6. However, there is a lack of detail concerning how the segments of dendrites were selected for study.

Are these apical or basal dendrites?

All experiments were conducted in dissociated primary hippocampal neuronal cultures, where it is not possible to differentiate apical from distal dendrites.

Proximal or distal?

Fig. 6 experiments were performed in distal dendrites, on average, 110 μm away from the cell body.

Was a single dendrite studied per neuron, or do the values reflect averages of multiple dendrites?

In Fig. 6, up to 3 spines and their corresponding dendrites were studied per neuron. Each spine was counted as a single experiment, and the spines were averaged to get the final data summary.

Were defined mitochondrial compartments used to select the segment to analyze?

Spines were chosen for plasticity induction by checking whether there was at least one mitochondrion within 3 μm from the base of the spine. However, the length of the mitochondrial compartment was not predefined.

Was the activated spine always proximal to the other spines studied as depicted in the schematic?

Moving distally, there is often a decrease in mitochondrial density and/or membrane potential - how was this controlled for?

The activated spine was not always proximal to the other spines. Once the plasticity-induced spine was analyzed for its spine-head width, the adjacent spines on either side of the plasticity-induced spine, within 45 μm distance, were used for analysis. However, to address the reviewer's question, we analyzed mitochondrial density within the same distance ranges of 0-15, 15-30, and 30-45 μm , in a dendritic segment 110 μm away from the cell body (see Methods) and did not find a statistically significant difference (Supplementary Fig. 6k).

All this additional information is now added to the Methods section under 'Spine structural plasticity measurements' and Supplementary Fig. 6k.

2. There is a disconnect between the length of the stable compartment defined morphologically, which drops to ~17 microns according to Fig. S6E, and the region of adjacent spines whose head size gains are proposed to be fueled by this compartment, as there are none in the 0-15 micron range. If compartment size is the determining factor in synchronizing spine head width responses in adjacent spines, one might expect at least a trend of spine head responses correlating with the now smaller compartment.

We have clarified this important point in the main text. Although in Supplementary Fig. 4a and Supplementary Fig. 4b, $t=5$ min (previously Fig. S6E), the mitochondrial compartment length in VAP KD is $17\ \mu\text{m}$ at the beginning of the photoactivation experiment, the compartment itself is unstable and reduces in length over the one-hour imaging duration to $8\ \mu\text{m}$ (Supplementary Fig. 4b, $t=60$ min). This destabilization is further confirmed by the compartment stability index measured in Fig. 4c, d, which shows that dendritic mitochondria in VAP KD are the most unstable compared to those in Snca KD and Srgap2 KD. In contrast, the Control mitochondrial compartment length stays the same, $\sim 30\ \mu\text{m}$, throughout the one-hour imaging duration (Supplementary Fig. 4a). We reason that this stabilization of the $\sim 30\ \mu\text{m}$ mitochondrial compartment is critical in determining the dendritic length fueled by mitochondria in adjacent spines (within $30\ \mu\text{m}$) in Fig. 6d-g. Since in VAP KD, the shortened $\sim 17\ \mu\text{m}$ mitochondrial compartment is unstable, they are unable to fuel the adjacent spines even within the $0\text{-}15\ \mu\text{m}$ range. We have now reemphasized this detail in the Results section.

3. Also, Supplementary Figure S6A shows a statistically significant increase in spine size at baseline in the VAP depletion group. However, if VAP helps to stabilize the mitochondria near spines, it is unclear why VAP depletion would lead to this increase. Further investigation is needed to understand the relationship between VAP, mitochondrial stability, and spine size regulation.

We want to clarify here that although Supplementary Fig. 6b (previously Figure S6A) shows a statistically significant increase in spine size, the $\Delta\text{spine size}_{\text{increase}}$ is very small = $0.1 \pm 0.03\ \mu\text{m}$ (control, $0.9 \pm 0.03\ \mu\text{m}$; VAP KO, $1.0 \pm 0.04\ \mu\text{m}$). This increase corresponds to a 1.1-fold increase in spine size and is comparable to the spine size fluctuations of Control, uninduced spines in Fig. 6c. We have clarified this point in the Results section now. Please also see our response to your related question 5 below.

4. Although the text states there are no basal differences in calcium influx between control and VAP KO, the graphs in Fig. 6B look strikingly different. More work is needed to understand the effects of VAP KO on spine head size and on calcium flux at baseline before considering activity-dependent changes. Is the assay already saturated at baseline for VAP KO?

Thank you for pointing this out. We want to clarify that there is indeed no difference in spine calcium influx between Control and VAP KO, which is clear in the representative trace in Supplementary Fig. 6f and averaged data in Supplementary Fig. 6h. In addition, we have now added data to show that there is also no difference in baseline spine calcium between Control and VAP KO (Supplementary Fig. 6g), and therefore the assay is not saturated for VAP KO. However, we note that the line profile that was initially shown in Fig. 6b was not representative of this result. Therefore, we have now replaced the line profile in Fig. 6b with one that is representative of the data in Supplementary Fig. 6g.

5. What percentage of the spines at different distances from the stimulated spine are increasing in size vs. decreasing in size. Is there a greater percentage of spines getting smaller due the larger average size of VAPKO spines at baseline?

We want to reemphasize here, as in response to question 3, that the average spine size increase is small ($0.1 \pm 0.03\ \mu\text{m}$) in VAP KO, even though statistically significant compared to Control.

However, we agree with the reviewer's concern that spines larger to begin with might not exhibit spine plasticity as much as smaller spines. While a greater percentage of spines get smaller in VAP KO, the key here is to characterize if the inability of spines to exhibit sustained plasticity in VAP KO is due to their larger baseline spine size or destabilized mitochondria. To rule out the contribution from larger baseline spine size, we only investigated a subset of Control and VAP KO spines in Supplementary Fig. 6b that are comparable in baseline spine size (Supplementary Fig. 6b dotted box, spine size $0.76 - 1.3\ \mu\text{m}$). We plotted the normalized spine-head width over time for the subset of these Control and VAP KO spines (Supplementary Fig. 6c). We find that even when the VAP KO spines are of similar initial baseline spine size as Control, they still do not exhibit sustained structural plasticity. This analysis

confirms that the inability of VAP KO spines to exhibit sustained structural plasticity is not due to their larger baseline spine size. We did a similar analysis for the adjacent uninduced spines and got similar findings (Supplementary Fig. 6d, e).

We have now added this explanation to the Results section. Similar questions were also raised by reviewer 3, question 3, and we have addressed them.

6. The potential roles of altered fission-fusion dynamics, mitochondrial turnover are not considered. VAP depletion results in the shortening and destabilization of mitochondrial compartments in dendrites. We agree with the reviewer that this shortening of mitochondrial compartments must be a result of altered fission-fusion balance in VAP KO compared to Control. Indeed, recent findings show that the additional role of VAP in lipid transfer between ER and mitochondria is important for mitochondrial fusion¹. Hence in the absence of VAP, it is arguable that mitochondrial fusion is affected, resulting in mitochondrial shortening. We have now added this sentence to the Discussion section.

Mitochondrial turnover, on the other hand, takes an average of 27 days². However, our VAP depletion experiments were done within 7 days of transfecting shRNA or sgRNA. Therefore, the role of mitochondrial turnover should be minimal in our experiments.

7. Is the inability to sustain increased spine size due to decreased mitochondrial content in general, and can this be replicated by other causes of energy depletion?

We thank the reviewer for this suggestion. Consistent with shortened mitochondrial length and compartment length in VAP KO neurons compared to Control (Supplementary Fig. 4a-c), we find that mitochondrial content (otherwise density) is also reduced in VAP KO (Supplementary Fig. 6k, see Methods).

Our findings are further supported by our earlier work, where we showed that local perturbation of a 30 μm mitochondrial compartment in a dendritic segment affects spine size increase and sustenance within that segment³. We have now added this explanation to the Results section.

In addition, in response to reviewers 2 and 3, questions 13 and 1, respectively, we measured mitochondrial calcium in VAP KO neurons and found that mitochondrial calcium influx is affected during synaptic plasticity (Supplementary Fig. 6j). As mitochondrial calcium is critical in activating enzymes of the Krebs cycle for ATP generation, we hypothesized that VAP KO affects local mitochondrial ATP generation due to reduced mitochondrial calcium influx. Using our newly developed mitochondria- and spine-specific ATP reporters, we also find that mitochondrial and spine ATP levels are reduced in VAP KO neurons during synaptic plasticity. However, since the two ATP reporters are not yet fully characterized and published, we have added only the mitochondrial calcium data to this manuscript. We, therefore, conclude that the shortened and destabilized mitochondria leading to low mitochondrial content, in addition to the reduced mitochondrial calcium influx and ATP generation, affect sustained synaptic plasticity in VAP KO. We have added this explanation to the Results and Discussion sections.

8. Does VAPA/B knockdown affect the percent of PSD95 or Homer positive spines? Could this account for the loss of co-stimulatory head changes in adjacent spines?

In Supplementary Fig. 6a, we show that the percent of PSD95- or Homer-positive spines per dendrite (spine density per 10 μm dendrite) does not change in VAP KO compared to Control. So, we do not think this is a factor in the observed loss of costimulatory head changes in adjacent spines.

9. There is a lot of variability in the numbers of neurons and animals used, ranging from 30 neurons from 10 animals in the supplementary figure to only 4 neurons from 2 animals in several key figures in the main manuscript. A more detailed description of the number of neurons, number of animals for each

group, and number of times the experiment was replicated is necessary to evaluate reproducibility. It would be helpful to color code the spines from different neurons or different animals in 6E, for example, to see if the variability in direction of spine size change from baseline segregates by animal or by neuron.

We have now added the number of spines/dendrites/axons/neurons and the number of animals for each of the conditions in all main figures and most supplementary figures. For a few supplementary figures, we have combined the n for all groups together due to the 350-word limit for figure legends. We have also color-coded the spines by neurons in Fig. 6f, g, Supplementary Fig. 6e, and do not see any segregation by neuron (or animal, data not shown). We also mention in the Methods section that no statistical method was used to predetermine sample size. Sample sizes were similar to or larger than those reported in previous publications in the field and sufficient for our claims based on statistical significance.

10. What is the p value for Control vs. MCU neg control in Fig. 3C
It is 0.3, and we have now added it to Fig. 3d (previously Fig. 3C).

11. In Fig. 6C, is 60 min significantly different from 2 min in the VAPKO curve?

We performed a paired sample t-test between the data points at t=2 min and 62 min in VAP KO, and the p-value is 0.06 for the normalized data and 0.04 for the unnormalized data. We have now added this detail in the figure legend. Also, see our response to reviewer 3, question 3.

12. Further validation of the APEX2 strategy is needed. The system may be too promiscuous if the majority of APEX-OMM labeled proteins are soluble proteins that quickly diffuse away. In Fig. 1B, it would be helpful to see what happens after permeabilization of the plasma membrane to allow labeled cytosolic proteins to diffuse out. One would expect to see a pattern similar to 1C, verifying that some labeled proteins are stably associated with the OMM.

We tried doing the proposed experiment where following APEX-OMM labeling, we permeabilized the neurons either with Triton-X or with Streptolysin-O to wash away the soluble proteins. However, despite testing various conditions with these two permeabilizing agents, either the permeabilization was too strong (in the case of Triton-X) that the neurons were not left intact, or the permeabilization was too mild (in the case of Streptolysin-O) that the soluble proteins could not be washed away effectively.

So, to address this question, we took a different approach. We compared our OMM proteome, with all the soluble proteins in the neuropil proteome⁴ and found only 31 (24%) overlapping proteins indicating that APEX-OMM labeling is not too promiscuous that it only results in soluble proteins. This data is now added to Supplementary Table 1, and we have added the explanation to the Results section.

13. It is curious that the screen failed to detect any proteins involved in axonal mitochondrial interactions with actin; for example, FHL2 mentioned in the text was not in the supplementary table (and as pointed out VAPB itself was not identified).

We thank the reviewer for pointing out this concern. Fhl2 is recruited to mitochondria in response to glucose influx when glucose is shifted from a low to a high concentration⁵. However, the APEX-OMM labeling was performed at uniform glucose concentrations in the absence of any induced glucose influx. We think the increased glucose influx requirement to recruit Fhl2 to axonal mitochondria, is why we do not see it in our OMM proteome list. Furthermore, the hippocampal neuronal protein, transcript and translome levels of Fhl2 are low compared to Vapa^{4,6}. We also note that Vapb protein, transcript and translome levels are lower than Vapa in hippocampal neurons (see below)^{4,6}. CaMKIIa values were added for better comparison, as it is a well-expressed protein in neurons. Please see similar question 16 addressed below. We have now added this explanation to the Discussion section.

CaMKIIa protein log2 iBAQ 32.2
Vapa protein log2 iBAQ 26.97
Vapb protein log2 iBAQ 25.75
Fhl2 protein log2 iBAQ 22.99

CaMKIIa transcriptome average expression 26370.07
Vapa transcriptome average expression 4915.09
Vapb transcriptome average expression 627.48
Fhl2 transcriptome average expression 384.93

CaMKIIa translome average expression 46467.88
Vapa translome average expression 1782.56
Vapb translome average expression 611.49
Fhl2 translome average expression 431.29

Can authors comment on what steps were taken to validate sensitivity for known OMM interacting proteins and specificity relating to proteins that may bump the OMM with no functional interaction?

We mention in the Discussion section that our hippocampal neuronal APEX-OMM proteome shows a 53% overlap with the published HEK cell APEX-OMM proteome (Supplementary Table 1, Hung et al., eLife 2017), which includes 7 of the 8 proteins we have investigated in our study. This observation validates the sensitivity of our method for known OMM interacting proteins.

To further confirm that our OMM proteome list is not merely proteins bumping OMM by chance with no functional interaction, we compared our OMM proteome list to an abundance-ranked soluble protein list from the neuropil obtained by the same methodological approach⁴ (Supplementary Table 1). While the digests were different (the samples in Biever et al.,⁴ were generated from lysed tissue, and our samples were produced via affinity purifications), the LC-MS was very similar in terms of chromatography columns, setups, and acquisition strategies.

We think that proteins that may bump the OMM with no functional interaction should also be abundant soluble proteins that statistically hit the OMM from time to time in non-directed diffusion. We find only 17% of our OMM proteome overlaps with the first 200 abundant soluble proteins, suggesting that we are not just sub-sampling the most abundant soluble, cytosolic proteins that randomly come into proximity via diffusion. Furthermore, our OMM proteome list is very different from the total soluble protein list (only 24% overlap), so we consider the likelihood of it happening as very low.

We have added these details to the Results section and updated Supplementary Table 1.

14. In Fig. S4E – both X(S)NCA and SRGAP2 show a greater effect on compartment length compared to VAP. What was rationale to focus on VAP? In Fig. 4, the SRGAP2 data may be underpowered to detect significant differences as this group has much fewer data points than the other groups.

Yes, we agree that in Supplementary Fig. 4a (previously Fig. S4E), Snca KD and Srgap2 KD show a larger decrease in mitochondrial compartment length compared to VAP KD. However, we want to bring your attention to Fig. 4, where VAP KD shows a statistically significant, reduced compartment stability index than Snca KD and Srgap2 KD, suggesting that the mitochondria in VAP KD dendrites are more unstable than in Snca and Srgap2 KD dendrites. Therefore, we focused on VAP as we were interested in candidates that destabilized the mitochondrial compartment and not just shortened the mitochondrial compartment. We agree with the reviewer that Srgap2 was underpowered in Fig. 4, so we added new data, and the result remains the same. We have now added this explanation to the Results and Discussion sections.

Minor points:

15. Since all 8 proteins had effects on actin-mitochondria contacts, what was rationale to focus only on VAP, SNCA and SRGAP2?

We started our secondary screen with VAP, Snca, and Srgap2, and we found a strong phenotype with VAP. So, we decided to focus on VAP for the rest of the study. We will be investigating the rest of the five candidates in the future.

16. Given differences in abundance that was raised as an explanation for why the APEX procedure did not identify VAPB, it would be interesting to see the relative abundance of VAPA and VAPB by western blot or RT-PCR in hippocampal cultures.

The Schuman lab has a published⁶ database that allows one to visualize the expression levels of different genes of interest at the transcriptome and the translome level in the rat hippocampal neurons: <https://public.brain.mpg.de/dashapps/localseq/info>. Since our APEX-OMM proteome measurements were done in the Schuman lab that used the same hippocampal slice and neuronal culture preparation protocol for the transcriptome/translation measurements, the relative gene expression levels should be comparable between our measurements. In addition, the neuropil proteome, also published by the Schuman lab, was obtained by the same methodological approach⁴ as our OMM proteome. While the digests were different (the samples in Biever et al.,⁴ were generated from lysed tissue, and our samples were produced via affinity purifications), the LC-MS was very similar in terms of chromatography columns, setups, and acquisition strategies.

From these datasets, we find that at the proteome, transcriptome, and translome levels, VAPA is highly expressed compared to VAPB, and FHL2 expression is even lower than VAPB. We have added CaMKIIa values for better comparison, as it is a well-expressed gene in neurons.

CaMKIIa protein log2 iBAQ 32.2

Vapa protein log2 iBAQ 26.97

Vapb protein log2 iBAQ 25.75

Fhl2 protein log2 iBAQ 22.99

CaMKIIa transcriptome average expression 26370.07

VAPA transcriptome average expression 4915.09

VAPB transcriptome average expression 627.48

FHL2 transcriptome average expression 384.93

CaMKIIa translome average expression 46467.88

VAPA translome average expression 1782.56

VAPB translome average expression 611.49

FHL2 translome average expression 431.29

We have added this explanation to the Results, Methods, and Discussion sections.

17. Authors should strive to utilize more consistent terminology that clearly indicates when both paralogs were knocked down or are being discussed together, such as VAPA/B. On line 143, it states that VAPA was knocked down, but line 190 it indicated that all experiments involved knockdown of both A and B, and in many places VAP is used.

Thank you for noting this inconsistency. On line 143, we actually meant VAP (a and b) and not just Vapa. We have now corrected it, added an explanation, and fixed any inconsistencies in the rest of the text.

18. In Fig. 5, are either of the VAP(s) associated with somatic mitochondria?

Both Vapa and Vapb endogenous immunostaining signal looks widely distributed throughout the neuronal soma, and it overlaps with the neuronal somatic mitochondria signal (Mito-DsRed, see Fig. below). However, neuronal somata are 10-20-fold thicker than dendrites and axons and have mitochondria packed into a reticulated network, compared to sparser mitochondria in dendrites and axons. So, it is not possible to do the correlation coefficient-based enrichment analysis of Vapa and Vapb near somatic mitochondria at the confocal resolution, as we did for dendritic and axonal mitochondria in Fig. 5 and Supplementary Fig. 5.

However, in response to Reviewer 2's question 6, we investigated the requirement of VAP for mitochondria-actin interaction in the neuronal soma. We find using high-resolution Airyscan confocal that mitochondria-actin interaction is unaffected by VAP depletion in the neuronal soma (Supplementary Fig. 3j). This result indicates that VAP does not interact with mitochondria in the soma for mitochondrial-actin tethering. So, the VAP-dependent mitochondria-actin tethering and stabilization is specific to dendrites. We have now added this explanation to the Results section.

19. The main conclusion, which is repeated verbatim in several positions in the paper “VAP functions as a spatial stabilizer that temporally sustains synaptic plasticity for up to ~60 min and as a spatial ruler that determines the ~30 μm spatial dendritic segment fueled by mitochondria during synaptic plasticity” is not actually investigated or supported by the data presented. As only the 60 min time point was examined, it is unclear whether an intermediate-sized ~17 micron compartment might sustain plasticity for a shorter period of time, or as mentioned in Point 2, shorten rather than abolish the region of responsive adjacent spines.

We are unsure what the reviewer means by ‘as only the 60 min time point was examined’.

In Fig. 6c, we monitor spine structural plasticity at various time points of 2, 12, 22, 32, 42, 52, and 62 min since plasticity induction. In Control, spine size enlargement is observed as early as 2 min post-plasticity induction and is sustained for up to 62 min. However, in the absence of VAP, spine size enlargement is only observed up to 12 min, following which it does not sustain. Based on this result, we conclude that ‘VAP functions as a spatial stabilizer that temporally sustains synaptic plasticity for up to 60 min’.

The second part of our conclusion, ‘...as a spatial ruler that determines the ~30 μm spatial dendritic segment fueled by mitochondria during synaptic plasticity’, is based on Fig. 6d-g. In addition, please see our response to point 2, where we explain that mitochondrial stability (and not just length) is a key factor in fueling the 30 μm dendritic segment during synaptic plasticity. In this regard, we agree with the reviewer that if there is an experimental manipulation that shortens mitochondrial compartment length, but retains mitochondrial stability, we can test if it can sustain plasticity within a shorter dendritic segment or for a shorter period of time. So Srgap2 would have been an ideal candidate to test this hypothesis as Srgap2 KD shortens mitochondria but does not destabilize it. However, Srgap2 KD neurons were not as active as Control, by cytosolic calcium transients, and their spines were immature, as has been reported before⁷. So, it was not possible to perform spine plasticity measurements in Srgap2 KD neurons. We also measured synaptic plasticity in Snca KD neurons that exhibit shortened mitochondria and show a trend towards mitochondrial destabilization that is not statistically significant. Snca KD neurons behaved similarly to VAP KO neurons, exhibiting defects in sustained synaptic plasticity, but it was not statistically significant (see below).

20. In the synaptic plasticity experiment, line 227 lists the early time point as 12 min. Is this a typo? Thank you for pointing it out. We meant t=2 and 12 min (and not t=0 and 12 min). We include t=12 min because spine sizes in VAP KO at both t=2 and 12 min are not statistically different from the Control in Fig. 6c. The typo is now corrected in the Results section.

Reviewer #2 (Remarks to the Author):

This manuscript describes a role for VAP in tethering dendritic mitochondria to actin, and a potential effect on dendritic spines. They first conduct APEX-OMM to identify proteins near neuronal mitochondria, and then examine the ones which are implicated in the cytoskeleton. They then examine how 8 of these including VAP regulate actin around mito, and show that VAP KD then decreases actin around mitochondria, and also mitochondrial fluorescence loss over time. They further show that VAPB is enriched in dendrites, and finally show that VAP KO cells have decreased spine size after stimulation. Overall, the concept of identifying actin-related proteins near mitochondria which regulate dendritic mitochondria and spine size is very interesting. However, there are currently major gaps in the mechanisms and conclusions, which could be greatly improved by addressing the comments below.

Major comments:

- Fig 2: The authors may want to check how proteins are classified (including their roles/localization). For example, SNCA is not classically thought of as a cytoskeletal protein. We want to clarify here that we do not mention that Snca is a cytoskeletal protein, but a cytoskeleton-interacting protein (specifically, actin-interacting protein), based on the interactome database, BioGRID^{8,9}.
- Fig 2: As VAPA is localized to ER, and a list of other ER-localized proteins found in their list would be helpful if the authors are interested in the role of the ER on mitochondria. Of the 129 proteins in the OMM proteome, 18 proteins are GO annotated as “Endoplasmic Reticulum”, and 3 proteins are GO annotated as both “Endoplasmic Reticulum” and “Mitochondria”. These proteins might be relevant to investigate ER-mitochondria contact sites and ER-dependent mitochondrial stabilization in neurons in the future. We have now added this information to Supplementary Table 1 and the Discussion section.
- Fig 3: Authors may want to use a control of their Fis1 plasmid: ex. Does change in mito size (ex. Drp1 KD, vs Mfn2 KD) alter the mito-actin interaction (%) measure? We thank the reviewer for suggesting this control experiment. We performed Drp1 KD to lengthen mitochondria; and Mff OE (Mitochondrial Fission Factor Over Expression) as it was readily available, instead of Mfn2 KD, both of which shorten mitochondria^{3,10}; and measured mito-actin interaction percentage. We found no difference in mitochondria-actin interaction percentage in both conditions compared to the Control. Therefore, this assay is insensitive to mitochondria size, and the data is now added to Supplementary Fig. 3c.

4. Fig 3: Images of normal Lifeact-mCherry with mitochondria, colocalized with their Fis1-Lifeact-GFP (or in conditions without it) would be helpful, to confirm that their new plasmid is not abnormally disrupting actin dynamics or its enrichment near mitochondria.

We had previously used F-actin-mCherry as an indicator of actin and showed that Fis1-Lifeact-GFP expression does not cause any abnormal aggregation of actin in dendrites in Fig. S3B. In response to this suggestion from the reviewer, we have replaced it with Supplementary Fig. 3a with actin signal measured using normal Lifeact-mCherry. Again, we find that Fis1-Lifeact-GFP expression does not abnormally disrupt actin or its enrichment near mitochondria.

In addition to the static images, we quantified spine actin dynamics, as a proxy measurement for general actin dynamics, in neurons expressing Fis1-Lifeact-GFP compared to control (absence of Fis1-Lifeact-GFP). We quantified spine actin dynamics over two minutes (the same duration as the plasticity induction protocol) using a previously established method to measure the average displacement of the weighted center of mass of spine actin¹¹. We found no difference in spine actin dynamics in neurons expressing Fis1-Lifeact-GFP compared to the Control (Supplementary Fig. 3b).

We have now added these details to the Results section and the new analysis method to the Methods section.

5. Fig 3: It is also unclear if VAP KD disrupts all actin, or just actin around mitochondria, so images of normal Lifeact throughout neurons upon either VAP KD or KO would be helpful.

We thank the reviewer for this suggestion and have now investigated actin using normal Lifeact-mCherry in Control, VAP KD, and VAP KO dendrites and axons. We do not see any actin disruption in neurons in VAP KD or KO compared to control. We now show representative dendrites, spines, and axons from these neurons in Supplementary Fig. 3e, f.

We also quantified spine actin dynamics (as explained above in response to question 4) and found no difference between Control and VAP KO neurons (Supplementary Fig. 3g).

6. Fig 3C: Does KD of candidate proteins such as VAP change mito-actin interaction (%) in cell bodies?

We thank the reviewer for raising this point. We investigated the requirement of VAP for mitochondria-actin interaction in the neuronal soma. We find using high-resolution Airyscan Confocal that mitochondria-actin interaction is not affected by VAP depletion in the neuronal soma (Supplementary Fig. 3j). This result indicates that VAP does not interact with mitochondria in the soma for mitochondrial-actin tethering. So, the VAP-dependent mitochondria-actin tethering and stabilization is specific to dendrites. We have now added this information to the Results section.

7. Fig S4: The authors should show KD efficiency by western blot (rather than by % decrease in immunofluorescence, which can be subject to changes in cell morphology, and secondary antibody fluorescent aggregates)

Transfection efficiency is really low in neuronal cultures (only 20-30 neurons are transfected in a dish plated with 60,000 – 80,000 neurons). So confirming and quantifying KD/KO efficiency on a neuron-to-neuron basis is the best way - it also allows us to investigate the same neuron in which we did live imaging, re-find it post-immunostaining, and confirm and quantify the KD/KO efficiency. This reinvestigation of the same individual neuron cannot be done by western blot. Moreover, given that western blot needs a lot of protein sample amount, neurons will have to be pooled from many different culture batches, and given that KD/KO transfection efficiency is really low per batch, the global protein intensity changes between control and KD/KD condition will be not representative of an individual neuron's phenotype. Besides, we do not see any obvious changes in cell morphology or secondary antibody aggregates in our immunostaining experiments, and this way of confirming KD/KO efficiency is accepted in the field^{3,12}. Moreover, the key here was to disrupt mitochondrial stabilization and reduce

the mitochondrial length, compartment length, and density, which we verified both by VAP KD and VAP KO (Fig. 4a-d, Supplementary Fig. 4a-d, 6k).

8. Fig 3: How does VAPA mechanistically tether actin to mitochondria? (Does it increase actin polymerization; or bind directly to actin; and /or mitochondria)?

VAP is identified as an actin interactor in the BioGRID database based on articles that studied the human interactome network^{13,14}. However, how VAP tethers actin to mitochondria is unclear and will be investigated in our future studies. It has been shown that VAP-dependent actin nucleation is required for endosome-ER contacts. So, a similar actin nucleation process might facilitate ER-mitochondria contacts. We have now added this detail to the Discussion section.

9. Fig 4A: It is unclear if the decrease in mito fluorescence in all conditions is from bleaching, or from fission/fusion? Can the authors confirm this a different way?

To address this question, we revisited our photoactivation timelapse images that had a few static mitochondrial regions that were photoactivated in the background. We measured the fluorescence of these regions over the same duration of imaging and found a negligible decrease. So, we conclude that the decrease in mitochondrial fluorescence is not from bleaching. This data is added to Fig. 4c.

10. Fig 5: These images are not very clear at this resolution. Showing perhaps a cytosolic and/or ER marker would be helpful to know the shape of the dendrites and axon (and to know whether VAPB is just filling the area or are just all over the ER, rather than specific to mito).

We now show the Map2 channel (blue) that was used as a dendritic fill to see the shape of the dendrites in Fig. 5b, e. It is clear that Vapb is not filling throughout the dendritic area but shows interspersed gaps similar to dendritic mitochondria. Unfortunately, we did not have an axonal fill in these experiments. However, the axonal shape can be traced with the help of the sparse Vapa/Vapb signal along the axonal length, and the traces are now added to the axonal images in Fig. 5c, f. We also want to clarify that the images in Fig. 5b-f are from straightened dendrites and axons, acquired across Z-stacks, and converted to a maximum intensity Z-projection. Therefore, the whole dendritic or axonal segment is projected onto the same imaging plane. So, an absence of Vapb signal denotes an absence of Vapb and is not due to that particular dendritic segment being out of focus from the imaging plane. This detail is now added to the Fig.5 legend and the Methods section.

In addition, we imaged ER along with mitochondria and Vapb. At confocal resolution, the ER signal looks like a cytosolic fill, while the Vapb signal is still enriched near the mitochondria and not just all over the ER (Supplementary Fig. 5f, g). This detail is now added to the Results section.

11. Fig S5: Can the authors show any live movies of VaPB-emerald with Mito-DsRed, to further show that the signals are moving together over time?

We thank the reviewer for this suggestion. We revisited the Vapb-emerald + Dsred-Mito representative images in Supplementary Fig. 5a, b. Indeed, we found that in dendrites, mitochondria that undergo fission and move away from the parent mitochondria are still enriched with the Vapb-emerald signal (Supplementary Fig. 5c, left). However, we did not see such an enrichment in moving axonal mitochondria (Supplementary Fig. 5c, right). We now mention it in the Results section.

12. Fig 6: Do VAP KO cells also show similar effects on photoactivation loss (similar to Vap KD in Fig 3)? Are there changes in mitochondrial or actin dynamics (in general using normal Lifeact) in these VAP KO cells?

We performed new experiments to measure the photoactivated compartment stability index in VAP KO dendrites and found the same result as in VAP KD dendrites. The mitochondrial compartments are unstable in VAP KO compared to those in Control (Supplementary Fig. 4d).

In addition, as mentioned in our response to question 5, we investigated actin using normal Lifeact-mCherry in control, VAP KO, and VAP KD dendrites and axons. We do not see any actin disruption in neurons in VAP KO or KD compared to Control. We now show representative dendrites, spines, and axons from these neurons (Supplementary Fig. 3e, f). We also quantified spine actin dynamics as a proxy measurement for general neuronal actin dynamics (as explained above in response to questions 4, and 5) and found no difference between Control and VAP KO neurons (Supplementary Fig. 3g).

13. Fig 6: What happens to mitochondrial dynamics/density/size/mitochondrial calcium and Fis1-lifeact in VAP KO cells upon 2-photon uncaging? Do changes in mitochondria and actin correspond to changes in spine shape (in both activated and neighboring spines) in VAP KO cells? This would help to confirm that changes in spines is due to VAP's role in mitochondria/actin interactions, rather than through its function on the ER.

We agree that it would be useful to measure mitochondrial dynamics via photoactivation upon 2-photon uncaging. Our experiment design requires Mito-PAGFP (green channel) to photoactivate a mitochondrial compartment and Mito-Dsred (red channel) to identify mitochondria to be photoactivated, as the non-photoactivated Mito-PAGFP signal is insufficient to identify mitochondria. Given that the green and red channels are taken, it is not possible to visualize spine stimulation, by either using GCaMP or RCaMP, to confirm 2-photon uncaging. Hence it is not possible to combine mitochondria photoactivation and 2p-uncaging in the same experiment.

We also agree that measuring Fis1-Lifeact dynamics in VAP KO upon 2p uncaging would be useful. However, our Airyscan confocal setup used to quantify mitochondria-actin interaction does not allow us to do 2-photon uncaging simultaneously. So, we imaged Fis-Lifeact in our spinning disc confocal system, where we do the 2-photon uncaging. However, this system does not provide the high resolution required to quantify mitochondria-actin interaction. So, we only imaged Fis-Lifeact fluorescence at the base of the plasticity-induced spines and did not find a consistent localization or dynamics of Fis1-Lifeact signal. It is possible that confocal imaging resolution is the limiting factor here. Therefore, these experiments will need to be performed with Airyscan or a super-resolution method combined with 2p-spine plasticity induction in the future. We now mention it in the Discussion section.

However, to address this question, we measured mitochondrial calcium in VAP KO neurons and found that mitochondrial calcium influx is reduced compared to Control during synaptic plasticity (Supplementary Fig. 6j). As mitochondrial calcium is critical in activating enzymes of the Krebs cycle for ATP generation, we hypothesized that VAP KO affects local mitochondrial ATP generation due to reduced mitochondrial calcium influx. Consistent with this data, using our newly developed mitochondria- and spine-specific ATP reporters, we also find that mitochondrial and spine ATP levels are reduced compared to Control in VAP KO neurons during synaptic plasticity. However, since the two ATP reporters are not yet fully characterized and published, we have added only the mitochondrial calcium data to the manuscript. We also measured ER calcium as a correlate for ER function in VAP KO neurons during synaptic plasticity, in response to reviewer 3 minor point 4, and found no effect compared to the Control (Supplementary Fig. 6i). We, therefore, conclude that the shortened and destabilized mitochondria leading to low mitochondrial density (Fig. 4a-d, Supplementary Fig. 4a-d, 6k) in addition to reduced mitochondrial calcium influx and ATP generation affect sustained synaptic plasticity in VAP KO. We have added this explanation to the Results, Discussion, and Methods sections.

14. Fig 6: Does VAP localization alter upon 2 photon uncaging?

We imaged Vapb (using Vapb-emerald) and spine calcium (using RCaMP) simultaneously to address this question. We also tried imaging mitochondria in addition. However, since the green and red channels were already taken, we could only use the blue channel to image mitochondria (using Mito-BFP). Unfortunately, the 405 nm laser used for Mito-BFP imaging prematurely uncaged the MNI-caged

glutamate in the imaging buffer resulting in neuronal death. So, we decided to skip imaging mitochondria and only imaged Vapb and spine calcium, reasoning that any change in Vapb localization with mitochondria would be visible on imaging Vapb alone. On spine stimulation using the same plasticity induction protocol as in Fig. 6, we did not see any obvious change in local Vapb localization, measured by Vapb fluorescence intensity, at the base of the plasticity-induced spines, within 4 μm diameter of the plasticity-induced spine (see Fig. below).

It is possible that confocal imaging does not provide the adequate resolution required to monitor any small changes in Vapb localization upon 2p uncaging. Therefore, these experiments will need to be done in the future at super-resolution, for e.g., using spt-PALM¹⁵. We now mention it in the Discussion section.

15. Can the authors also see if specific mutations in VAP (that might disrupt its function or ER localization) alter this ability to disrupt spines?

The well-studied VAP mutation that disrupts ER-mitochondria association is Vapb-P56S¹⁶. Expression of Vapb-P56S in neurons, however, does not destabilize mitochondria in photoactivation experiments, as endogenous Vapb potentially outcompetes the mutant Vapb at ER-mitochondria contact sites. So, we used VAP KO as background and then expressed Vapb-P56S, or WT Vapb as control. However, the control WT Vapb-expressing neurons (in VAP KO background) still showed synaptic plasticity defects, indicating that it could not recover the KO phenotype. The Vapb-P56S-expressing neurons (in VAP KO background) also showed a similar phenotype as WT Vapb-expressing neurons (in VAP KO background). In the absence of a reliable control, this result is hard to interpret. We, therefore, mention in the Discussion section that future studies with the Vapb-P56S mutant mouse model will be able to address this question better.

Moreover, in response to reviewer 3, minor point 4, we investigated the role of ER in VAP KO neurons. We measured ER calcium as a correlate for ER function during synaptic plasticity, and found no effect in ER calcium release in VAP KO neurons compared to Control (Supplementary Fig. 6i). So, we conclude that the synaptic plasticity phenotype we observe in VAP KO arises from mitochondrial dysfunction and not ER dysfunction.

16. Multiple other proteins have previously implicated in the role of actin around mitochondria (ex. INF2: Korobova, Science 2013; Myosin II: Korobova, Curr Biol 2014; Spire1C: Manor, Elife 2015; Myo19: Coscia, J Cell Sci 2023). Do these proteins show up in the OMM-APEX? Can the authors see if VAP alters actin around mitochondria through changes in these proteins (such as altering their localization/dynamics/levels; or possibly, if KO of these proteins also changes their Fis1 actin probe, as a control)?

The four proteins the reviewer has listed, Inf2, Myosin II, Spire 1C, and Myo 19, are important for mitochondrial fission in non-neuronal cells. However, we did not find these proteins in our APEX-OMM proteome. Given that our APEX labeling is only for 1 minute, this duration is probably not enough to

capture mitochondria-associated fission proteins. It could also be due to a small fraction of mitochondria undergoing fission during the 1-minute duration of the OMM proteome labeling.

The other reason could be due to an overall low expression of these four proteins in neuronal dendrites, in contrast to non-neuronal cells. For example, three of the four proteins Myo II, Myo 19, and Spire 1C are not found in the published datasets for neuropil proteome, transcriptome, and translome^{4,6} that were collected from similar samples as our OMM proteome (see above). This observation suggests that these three proteins are either low in amount or absent in the neuropil. While Inf2 was found in the neuropil proteome, transcriptome, and translome lists, it was at low expression values compared to CaMKIIa, a well-expressed neuropil protein.

CaMKIIa translome 46467.88
Inf2 translome 864.7

CaMKIIa transcriptome 26370.07
Inf2 transcriptome 479

CaMKIIa protein log₂ iBAQ 32.2
Inf2 protein log₂ iBAQ 21.07

Nevertheless, we knocked down two of the four proteins Inf2 and Spire 1C, in neurons and found that the mitochondrial-actin interaction percentage was unaffected in dendrites (Supplementary Fig. 3c). We have now added this detail to the Results section.

Minor comments:

1. Fig 1: It is unclear why the APEX-matrix is being shown. Was mass-spec done for hits on APEX-matrix as well, and if so, how are these different from APEX-OMM?

APEX-matrix was only used as a control to show the difference in compartment-specific labeling between APEX-OMM and APEX-matrix. We did not perform mass spec analysis of APEX-matrix. We have now clarified this information in the Results section.

2. Fig 3A: List schematic as its own panel (not on the figure, which is a little hard to read).
Done. See new Fig. 3a.

3. Fig 3B: Zoom in examples would be extremely helpful.
Done. See updated Fig. 3b, c.

4. Fig 4A: Kymographs need a scale bar for time (y axis); and also a marker indicating time of initial photoactivation (if shown on kymograph).
Done. See updated Fig. 4b (previously 4A).

5. Fig 5A/B: Gray vs dotted gray arrows are a little hard to tell apart: perhaps use of different colors or types of arrowheads would be better.
Done. See updated Fig. 5a, d (previously 5A, B).

Reviewer #3 (Remarks to the Author):

In this article, Bapat and colleagues investigate how mitochondria are tethered to the cytoskeleton and how this tethering in dendrites may be involved in synaptic plasticity. Using a range of molecular and imaging techniques they generate a biotinylation screening and identify several candidate proteins that tether dendritic mitochondria. Next, they investigate the role of vesicle-associated membrane

associated protein (VAP) in more detail comparing axons and dendrites. They find that VAPB (but not VAPA) tethers mitochondria to the cytoskeleton in dendrites but not axons. Moreover, knocking out VAP impairs spine dynamics in protocols for inducing long-term synaptic plasticity. The molecular techniques are excellent and address an important research question, as it is becoming increasingly clear that mitochondria behave differently in various subcompartments of the cell. While most data are clearly described some of the conclusions lack supporting evidence.

Major points

1. The main conclusion the authors draw is that mitochondria ‘fuel’ synaptic plasticity, and that disrupting their tethering impairs their ability to provide sufficient ATP to support spine growth. This is alluded to in the introduction (line 65), in the paragraph title (line 205), the title of Figure 6 and in the manuscript title. However, there is no supporting data presented. Which fueling mechanisms are affected by the VAP KD/KO? Do mitochondria have a smaller ‘range’ (due to a smaller compartment, Fig. 6F), become more motile, is ATP production disrupted, or is there a role for mitochondrial calcium buffering? To examine fueling a critical piece of evidence is showing ATP production in control and VAP KO conditions, particularly as the authors point out in the discussion (line 326) that ER-MCSs are important for ATP production, and VAP is involved in ER-mitochondrial contact. This could be done by targeting an ATP sensor to mitochondria.

To address this question, we measured mitochondrial calcium in VAP KO neurons and found that mitochondrial calcium influx is reduced compared to control during synaptic plasticity (Supplementary Fig. 6j). As mitochondrial calcium is critical in activating enzymes of the Krebs cycle for ATP generation, we hypothesized that VAP KO affects local mitochondrial ATP generation due to reduced mitochondrial calcium influx. Consistent with this data, using our newly developed mitochondria- and spine-specific ATP reporters, we also find that mitochondrial and spine ATP levels are reduced in VAP KO neurons compared to Control during synaptic plasticity. However, since the two ATP reporters are not yet fully characterized and published, we have added only the mitochondrial calcium data to the manuscript. We also measured ER calcium as a correlate for ER function, in response to your minor point 4, and found no effect in ER calcium release in VAP KO compared to Control during synaptic plasticity (Supplementary Fig. 6i). We, therefore, conclude that the shortened and destabilized mitochondria leading to low mitochondrial density (Fig. 4a-d, Supplementary Fig. 4a-d, 6k), in addition to reduced mitochondrial calcium influx and ATP generation affect sustained synaptic plasticity in VAP KO. We have added this explanation to the Results, Discussion, and Methods sections.

2. The authors imply that mitochondria are smaller upon VAP KO/KD (Figure 6F), but no evidence is presented. MitoTracker or mito-dsRed imaging is required to provide a more direct readout of mitochondria. These data are also essential to combine with the photoactivation and spine uncaging experiments, showing overlays of images, at high resolution. How do mitochondria position relative to the targeted spines? This data will be important to interpret the 30 μ m range the authors refer to.

We want to clarify that the mitochondrial compartment length measurement we have provided in this manuscript is proportional to mitochondrial length. Mitochondrial compartment length represents the functional continuity of mitochondria measured based on the exchange of fluorescent or photobleached proteins within the compartment (see Methods)³. Whereas mitochondrial length is a physical parameter obtained by measuring mitochondria from end-to-end (see Methods). We have shown evidence through various images and analyzed, summarized data that mitochondrial compartment length is reduced in VAP KD and KO dendrites (Fig. 3b, 4a, b, 6d, e, Supplementary Fig. 3e, 4a, b). To further address the reviewer’s question, we measured mitochondrial length, which also shows a reduction in VAP KO dendrites compared to the Control (Supplementary Fig. 4c, see Methods). In addition, we have added mitochondria images (measured using Dsred-Mito) to photoactivation experiments in Fig. 4a and spine uncaging experiments in Fig. 6d, e. All this information is now added to the Results, Discussion, and Methods sections.

3. In Figure 6B the GCaMP fluorescence at baseline seems to be higher in the VAP KO. What's the explanation?

We want to clarify that there is indeed no difference in baseline spine calcium between control and VAP KO neurons, which is evident in the new summarized data in Supplementary Fig. 6g. However, we note that the line profile that was initially shown in Fig. 6b was not representative of this result. Therefore, we have replaced it with a better representative of the summarized data in Supplementary Fig. 6g.

Is the baseline spine morphology affected? The authors write that VAP KO resulted in a “modest and negligible” spine-head size increase (line 219). However, data in Figure S6A demonstrates a significant increase (two-sample t-test, $n = 27$ from 7 animals). Referring to statistically significant differences as “negligible” seems inappropriate.

We want to clarify here that although Supplementary Fig. 6b (previously Fig. S6A) shows a statistically significant increase in spine size, the $\Delta\text{spine size}_{\text{increase}}$ is very small = $0.1 \pm 0.03 \mu\text{m}$ (spine head width: control, $0.9 \pm 0.03 \mu\text{m}$; VAP KO, $1.0 \pm 0.04 \mu\text{m}$). This increase corresponds to a 1.1-fold increase in spine size and is within the range of spine size fluctuations observed in Control, uninduced spines in Fig. 6c. We have clarified this point in the Results section now and have eliminated the term ‘negligible’ in describing this data.

The VAP KO shows several large spines $>1.2 \mu\text{m}$ not observed in control raising the question whether this limits their capacity to grow during the plasticity paradigm, explaining the limited spatial gradient of plasticity.

We agree with the reviewer's concern that spines larger to begin with might not exhibit spine plasticity as much as smaller spines. To test if the inability of spines to exhibit sustained plasticity in VAP KO is due to their larger baseline spine size or destabilized mitochondria, we investigated a subset of Control and VAP KO spines in Supplementary Fig. 6b that are comparable in baseline spine size (Supplementary Fig. 6b dotted box, spine size $0.76 - 1.3 \mu\text{m}$). We then plotted the normalized spine-head width over time for the subset of these Control and VAP KO spines (Supplementary Fig. 6c). We find that even when the VAP KO spines are of similar initial baseline spine size as the Control, they still do not exhibit sustained structural plasticity. This analysis confirms that the inability of VAP KO spines to exhibit sustained structural plasticity is not due to their larger baseline spine size. We did a similar analysis for the adjacent uninduced spines and got similar results (Supplementary Fig. 6d, e). We have now added an explanation in the Results section. Similar questions were also raised by reviewer 1, questions 3 and 5, and we have addressed them.

It would also be useful to see the absolute (non-normalized) data from the plasticity experiments in Figure 6.

The unnormalized data show the same result as the normalized data, where VAP KO spines are unable to sustain spine structural plasticity over 60 min, compared to Control spines (see below).

4. Figure 5D is missing a critical comparison to support the author's conclusion that VAPB is enriched near dendritic but not axonal mitochondria. It compares VAPB against VAPA but we need statistical comparison of dendritic vs. axonal VAPB.

We have now added the statistics for the correlation coefficient between VAPB (B and B_{live}) and mitochondria in dendrites compared with axons, and it is statistically significant (Fig. 5i, previously Fig. 5D). We have added the corresponding text to the figure legend.

Minor points

– In general, the figures could be increased in clarity. Merged images are lacking, for example, Figure 1D/E, Figure 5A/B (bottom). Conversely, separate images for each channel should be added to Figure 3A/B. Also, even though it was published before, it would be useful to add a small cartoon of the method used in Figure 4 or at least explain it briefly in the main text. Figure 4A would also become clearer if the fields of view to which the kymographs correspond were added (like Figure 3A), ideally at $t = 0$ and $t = 60$ minutes. The line profiles in Figure 1D and E are missing a scale along the x axis. Figure 5A/B would benefit from line profiles like the ones used in Figure 1 and Figure S5.

Done. See updated Fig. 1c, 1e, 3b, 3c, 4a, 4b, 5b-f, Supplementary Fig. 5a-c and Results section.

– In Figure 4C SRGAP2 doesn't show significant difference from control, but also has relatively few datapoints. What is the power? If more data points were added this could show that SRGAP2 is also important (which the authors hint to in the paragraph title). Please add data or clarify.

We have now added more data for Srgap2 KD in Fig. 4c, d, and the result remains the same - mitochondrial compartments were stable as in Control.

We also mention in the Methods section that no statistical method was used to predetermine sample size. Sample sizes were similar to or larger than those reported in previous publications in the field and sufficient for our claims based on statistical significance.

– In Figure S2B the authors claim an equal amount of protein input was used, although the input for "APEX-OMM" is higher in comparison to "– APEX". Quantifications are needed. This part also needs to be included in the main figures, as it is critical evidence. Independent and better evidence for the localization of APEX to the IMM and OMM could be provided, for example using ultrastructure electron microscopy.

We apologize for this oversight. We have now replaced the Coomassie experiment with the one where the input signal is comparable between APEX-OMM and -APEX. The biotinylation western blot has also been replaced with the one from the same batch as the Coomassie experiment, and the quantifications are added. See updated Supplementary Fig. 2a, b.

We, however, disagree that these data should be in the main figure and that independent evidence is needed to confirm the localization of APEX to the OMM (we did not use APEX targeted to the IMM). The two APEX constructs we used in this manuscript (APEX-OMM and APEX-matrix) are well characterized for their localization using fluorescence and electron microscopy by the Ting lab, who developed this method. We have, therefore, added the relevant literature to the Results section. What is critical, however, is to confirm if the APEX-OMM strategy is sensitive in identifying proteins interacting with the OMM. Therefore, we have compared our APEX-OMM proteome measured in neurons to the one published in HEK cells from the Ting Lab, and it shows a 53% overlap. Furthermore, we confirmed that the OMM interacting proteins we identified are not merely highly abundant soluble proteins bumping into the OMM. Please see our response to Reviewer 1, question 13, for further details. We have added these details to the Results section and updated Supplementary Table 1.

– It is unclear if ER function is still normal after VAP disruption. ER also is involved in synaptic plasticity

(see e.g. doi.org/10.1073/pnas.0905110106; doi: 10.1038/s41467-020-18889-5) and VAP is important for ER function as the authors point out. The authors should speculate on this or at least mention this caveat in the discussion.

We measured ER calcium as a correlate for ER function in VAP KO neurons during synaptic plasticity, and found no effect in ER calcium release compared to Control (Supplementary Fig. 6i). We have now added this explanation to the Results and Methods sections.

– In Figure 6 supporting images at high resolution across a longer range of the dendrite is missing. These are shown in Fig. S6C but should be included in the main figure.

Done. See updated Fig. 6d, e.

– The authors should better clarify the relationship of VAP with ALS or neurodegenerative diseases. They refer to ALS throughout the manuscript, even within the abstract (line 20) but don't make clear what the role of VAP and dendritic mitochondria tethering could have in these disorders.

We have now explained this better in the Discussion section.

– It is unclear which type of spines the authors included in their analysis (mushroom, stubby, ...). This should be clarified in the methods.

All PSD95-mCherry-positive or Homer2-mOrange-positive spines were used for analysis. We did not differentiate between mushroom, stubby, or any other spine shape. We have now added this detail in the Methods section.

– In Fig. S5A the background of the dendrite channel seems very low (almost absent) compared to that of the axon in Fig. S5B. Was the same background subtraction used? Dendritic mitochondria are much longer than those in axons. Could this not explain why the authors find more colocalization with VAPB in dendrites compared to axons?

In Supplementary Fig. 5a, b, we did not do any background subtraction. However, in Supplementary Fig. 5b, as the Vapb signal was too low in axons, the brightness/contrast was adjusted for better visualization, which also made the background noise more visible. This detail is now added to the figure legend.

It is a valid concern that as dendritic mitochondria are longer than axonal mitochondria, it might result in more Vapb colocalization near dendritic mitochondria than in axonal mitochondria. However, we do not see the same trend with Vapa, where it is equally enriched near dendritic and axonal mitochondria. So, the difference in enrichment between dendritic and axonal mitochondria seems to be unique to Vapb. We have now added this explanation to the Results section.

– Line 585, line 806 “neurons were replaced with ...”, please rewrite.

Done.

References

- 1 Subra, M. *et al.* VAP-A intrinsically disordered regions enable versatile tethering at membrane contact sites. *Dev Cell* **58**, 121-138 e129, doi:10.1016/j.devcel.2022.12.010 (2023).
- 2 Menzies, R. A. & Gold, P. H. The turnover of mitochondria in a variety of tissues of young adult and aged rats. *J Biol Chem* **246**, 2425-2429 (1971).
- 3 Rangaraju, V., Lauterbach, M. & Schuman, E. M. Spatially Stable Mitochondrial Compartments Fuel Local Translation during Plasticity. *Cell* **176**, 73-84 e15, doi:10.1016/j.cell.2018.12.013 (2019).
- 4 Biever, A. *et al.* Monosomes actively translate synaptic mRNAs in neuronal processes. *Science* **367**, doi:10.1126/science.aay4991 (2020).
- 5 Basu, H. *et al.* FHL2 anchors mitochondria to actin and adapts mitochondrial dynamics to glucose supply. *The Journal of cell biology* **220**, doi:10.1083/jcb.201912077 (2021).
- 6 Glock, C. *et al.* The translome of neuronal cell bodies, dendrites, and axons. *Proceedings of the National Academy of Sciences of the United States of America* **118**, doi:10.1073/pnas.2113929118 (2021).

- 7 Charrier, C. *et al.* Inhibition of SRGAP2 function by its human-specific paralogs induces neoteny during spine maturation. *Cell* **149**, 923-935, doi:10.1016/j.cell.2012.03.034 (2012).
- 8 McFarland, M. A., Ellis, C. E., Markey, S. P. & Nussbaum, R. L. Proteomics analysis identifies phosphorylation-dependent alpha-synuclein protein interactions. *Molecular & cellular proteomics : MCP* **7**, 2123-2137, doi:10.1074/mcp.M800116-MCP200 (2008).
- 9 Oughtred, R. *et al.* The BioGRID database: A comprehensive biomedical resource of curated protein, genetic, and chemical interactions. *Protein Sci* **30**, 187-200, doi:10.1002/pro.3978 (2021).
- 10 Toyama, E. Q. *et al.* Metabolism. AMP-activated protein kinase mediates mitochondrial fission in response to energy stress. *Science* **351**, 275-281, doi:10.1126/science.aab4138 (2016).
- 11 Tarnok, K. *et al.* A new tool for the quantitative analysis of dendritic filopodial motility. *Cytometry A* **87**, 89-96, doi:10.1002/cyto.a.22569 (2015).
- 12 Rangaraju, V., Calloway, N. & Ryan, T. A. Activity-driven local ATP synthesis is required for synaptic function. *Cell* **156**, 825-835, doi:10.1016/j.cell.2013.12.042 (2014).
- 13 Hein, M. Y. *et al.* A human interactome in three quantitative dimensions organized by stoichiometries and abundances. *Cell* **163**, 712-723, doi:10.1016/j.cell.2015.09.053 (2015).
- 14 Huttlin, E. L. *et al.* The BioPlex Network: A Systematic Exploration of the Human Interactome. *Cell* **162**, 425-440, doi:10.1016/j.cell.2015.06.043 (2015).
- 15 Obara, C. J. *et al.* Motion of single molecular tethers reveals dynamic subdomains at ER-mitochondria contact sites. *bioRxiv*, 2022.2009.2003.505525, doi:10.1101/2022.09.03.505525 (2022).
- 16 Nishimura, A. L. *et al.* A mutation in the vesicle-trafficking protein VAPB causes late-onset spinal muscular atrophy and amyotrophic lateral sclerosis. *Am J Hum Genet* **75**, 822-831, doi:10.1086/425287 (2004).

REVIEWER COMMENTS

Reviewer #2 (Remarks to the Author):

The authors have responded to the prior comments with several clarification points as well as with several important control experiments now shown in Main or Supplemental Figures, which help to strengthen their initial manuscript. I have no additional comments.

Reviewer #3 (Remarks to the Author):

In this manuscript NCOMMS-23-05538A Bapat and colleagues have clarified many of the previous concerns and significantly improved presentation of the findings. Nonetheless, there are a few points remaining.

The previous main concern (#1) was the lack of empirical support for 'fueling'. This concept is used throughout the manuscript, including the title. A similar question was asked by Rev #2, their point #7. The new Ca²⁺ imaging data shows that mitochondrial Ca²⁺ uptake is reduced in the VAP KO (Suppl. Fig 6j). In the rebuttal the authors claim that spine ATP levels are reduced based on new experiments with two sensors which "are not yet fully characterized and published". Stating results without numbers and only within a rebuttal obviously cannot be accepted as evidence. Thus, whether ATP synthesis in the VAP KO mitochondria is mechanistically involved in regulating spatially spine plasticity remains unclear. If the authors decide not to include the new ATP imaging the authors should refrain from writing 'fueling' and replace this with 'regulating' or 'controlling'. The idea that mt-Ca²⁺ is sufficient to assume ATP synthesis changes is unwarranted. Mt-Ca²⁺ and ATP synthesis are not always correlated (e.g. see 10.1038/s41467-018-07416-2 where increased mt-Ca²⁺ influx was not correlated with ATP synthesis).

Minor points.

1. The new Figure 4 has improved greatly with the added example timelapse and Mito-DsRed images. However, what remains unclear is why a reduction in photoactivated compartment fluorescence is interpreted as 'reduced compartment stability'. An alternative explanation could be there is more continuity between mitochondria, and that the photoactivated GFP spreads over a greater area? Please clarify. Furthermore, what explains the fission of PAGFP, e.g. at t = 55 min? Please clarify.

2. Figure 6. What explains the 30 μm ?

3. The authors respond to the previous main concern #3 with a clear explanation about the difference in spine size between the control and VAP KO group (it is in range of control spine size fluctuations), and that this explanation has now been added to the main text. However, it was missing in the main text. Please add.

The main text reads "The structural plasticity defect in VAP KO spines is not due to the slight increase in baseline spine size in VAP KO compared to Control, nor is it due to any deficiencies in baseline spine calcium, spine calcium influx, or ER calcium release at the base of the plasticity-induced spine (Supplementary Fig. 6bi)." That is more a statement and is unclear for the generally interested reader.

We thank the reviewers for reviewing our revised manuscript and for their positive feedback. We have added our point-to-point response to reviewer 3 to address their additional concerns.

Reviewer #2 (Remarks to the Author):

The authors have responded to the prior comments with several clarification points as well as with several important control experiments now shown in Main or Supplemental Figures, which help to strengthen their initial manuscript. I have no additional comments.

Reviewer #3 (Remarks to the Author):

In this manuscript NCOMMS-23-05538A Bapat and colleagues have clarified many of the previous concerns and significantly improved presentation of the findings. Nonetheless, there are a few points remaining.

1. The previous main concern (#1) was the lack of empirical support for ‘fueling’. This concept is used throughout the manuscript, including the title. A similar question was asked by Rev #2, their point #7. The new Ca²⁺ imaging data shows that mitochondrial Ca²⁺ uptake is reduced in the VAP KO (Suppl. Fig 6j). In the rebuttal the authors claim that spine ATP levels are reduced based on new experiments with two sensors which “are not yet fully characterized and published”. Stating results without numbers and only within a rebuttal obviously cannot be accepted as evidence. Thus, whether ATP synthesis in the VAP KO mitochondria is mechanistically involved in regulating spatially spine plasticity remains unclear. If the authors decide not to include the new ATP imaging the authors should refrain from writing ‘fueling’ and replace this with ‘regulating’ or ‘controlling’. The idea that mt-Ca²⁺ is sufficient to assume ATP synthesis changes is unwarranted. Mt-Ca²⁺ and ATP synthesis are not always correlated (e.g. see 10.1038/s41467-018-07416-2 where increased mt-Ca²⁺ influx was not correlated with ATP synthesis).

We agree with the reviewer’s concern and, therefore, have replaced the word ‘fueling’ with ‘supporting’ in the title and the rest of the manuscript.

Minor points.

1. The new Figure 4 has improved greatly with the added example timelapse and Mito-DsRed images. However, what remains unclear is why a reduction in photoactivated compartment fluorescence is interpreted as ‘reduced compartment stability’. An alternative explanation could be there is more continuity between mitochondria, and that the photoactivated GFP spreads over a greater area? Please clarify.

We agree with the reviewer that a reduction in photoactivated compartment fluorescence could be due to (i) more continuity between mitochondria due to fusion resulting in photoactivated GFP spreading over a greater area or (ii) destabilization of mitochondrial compartments. Regarding the former possibility, we mention in lines 363-366 that ‘Mitochondria in Control dendrites showed a modest decrease in photoactivated compartment fluorescence 60 minutes post-photoactivation, perhaps corresponding to fluorescent protein leak from mitochondria during basal-level mitochondrial dynamics such as fission and fusion...’. However, this decrease is minimal in Control (Fig. 4b, c, d). While there is a possibility that the larger decrease in photoactivated fluorescence in VAP KD is due to increased mitochondrial fusion and photoactivated GFP spreading to a larger area, our mitochondrial length and compartment length measurements show otherwise, that mitochondria are shorter in VAP KD/KO perhaps due to decreased fusion or increased fission (Fig. 3c, 4a, 6e, Supplementary Fig. 3e, 4a, b, c)

(also see Discussion lines 933-935). Furthermore, mitochondrial destabilization in VAP KD compared to Control is clear in our kymographs (representative Fig. 4b). Hence, we interpret the reduced photoactivated compartment fluorescence as reduced compartment stability. We have now added this additional explanation in the Figure legend.

Furthermore, what explains the fission of PAGFP, e.g. at $t = 55$ min? Please clarify.

We assume the reviewer is referring to the Control condition at $t=55$ min in Fig. 4a. Yes, fission and fusion do occur in Control mitochondrial compartments. Please refer to lines 363-366. However, it seems this balance is affected in VAP KD, resulting in shortened mitochondrial compartments (Fig. 3c, 4a, b, 6e, Supplementary Fig. 3e, 4a, b, c). Also, see Discussion lines 933-935, where we mention that the shortened mitochondrial compartments in VAP KD and KO could be due to reduced lipid transfer between ER and mitochondria and therefore reduced fusion.

2. Figure 6. What explains the $30 \mu\text{m}$?

We mention in lines 591-592 that 'Mitochondrial compartments are stable for ~ 60 min and are essential to fuel protein synthesis-dependent synaptic plasticity within $30 \mu\text{m}$ dendritic segments⁶'. We, therefore, hypothesized that if VAP is essential for mitochondrial stabilization, it will also determine the $30 \mu\text{m}$ dendritic segment supported during synaptic plasticity, see **updated lines** 594-595, 'absence of VAP will affect both the temporal sustenance of synaptic plasticity and the $30 \mu\text{m}$ dendritic segment exhibiting synaptic plasticity'. To test this idea, we analyzed spines adjacent to plasticity-induced spines within and beyond $30 \mu\text{m}$ distance. See **updated lines** 770-771, 'Furthermore, to determine the spatial dendritic segment supported by VAP-dependent mitochondrial stabilization, we investigated...'. We mention in lines 773-778 that 'In Control neurons, spines adjacent to the plasticity-induced spine within $0-15 \mu\text{m}$ showed structural plasticity consistent with earlier observations¹⁹⁻²⁴, with a similar trend in spines within $15-30 \mu\text{m}$ distance and no apparent structural plasticity in spines within $30-45 \mu\text{m}$ distance, at $t=2$ and 62 min post-plasticity induction (Fig. 6d-g). However, on VAP deletion, the ability of the adjacent spines to exhibit structural plasticity was affected as early as $t=2$ minutes post-plasticity induction (Fig. 6d-g)'. These data therefore suggest that the $30 \mu\text{m}$ sized mitochondrial compartments determine the $30 \mu\text{m}$ sized dendritic segment supported during clustered synaptic plasticity. When the $30 \mu\text{m}$ mitochondrial compartment is shortened and destabilized in VAP KO it also affects the size of the dendritic segment exhibiting clustered synaptic plasticity (see lines 780-783).

3. The authors respond to the previous main concern #3 with a clear explanation about the difference in spine size between the control and VAP KO group (it is in range of control spine size fluctuations), and that this explanation has now been added to the main text. However, it was missing in the main text. Please add.

The main text reads "The structural plasticity defect in VAP KO spines is not due to the slight increase in baseline spine size in VAP KO compared to Control, nor is it due to any deficiencies in baseline spine calcium, spine calcium influx, or ER calcium release at the base of the plasticity-induced spine (Supplementary Fig. 6bi)." That is more a statement and is unclear for the generally interested reader.

Done. See updated lines 743-746.